# Enhancing Small Medical Learners with Privacy-preserving Contextual Prompting

**Xinlu Zhang[1]\*, Shiyang Li[1], Xianjun Yang[1], Chenxin Tian[2], Yao Qin[1], Linda Ruth Petzold [1]**
[1]University of California, Santa Barbara
[2]Chinese Academy of Medical Sciences and Peking Union Medical College

## Abstract

Large language models (LLMs) demonstrate remarkable medical expertise, but data privacy concerns impede their direct use in healthcare environments. Although offering improved data privacy protection, domain-specific small language models (SLMs) often underperform LLMs, emphasizing the need for methods that reduce this performance gap while alleviating privacy concerns. In this paper, we present a simple yet effective method that harnesses LLMs' medical proficiency to boost SLM performance in medical tasks under *privacy-restricted* scenarios. Specifically, we mitigate patient privacy issues by extracting keywords from medical data and prompting the LLM to generate a medical knowledge-intensive context by simulating clinicians' thought processes. This context serves as additional input for SLMs, augmenting their decision-making capabilities. Our method significantly enhances performance in both few-shot and full training settings across three medical knowledge-intensive tasks, achieving up to a 22.57% increase in absolute accuracy compared to SLM fine-tuning without context, and sets new state-of-the-art results in two medical tasks within privacy-restricted scenarios. Further out-of-domain testing and experiments in two general domain datasets showcase its generalizability and broad applicability. Our code can be found at `https://github.com/XZhang97666/PrivacyBoost-SLM`.

Figure 1: **Synthetic medical data for illustration.** Though rich in domain-specific knowledge, medical data contains sensitive private information. We extract keywords to mitigate privacy concerns.

Steven Smith is a 60-year-old man admitted at Auckland Hospital. He was attended by Dr. Edward Jones at Date: 06/01/2008 . He has a past medical history significant for uncontrolled HTN who presents with a non-reducible right inguinal hernia. Patient first noticed a right sided bulge in 3 months prior. Every day it slips out and he has to manually push it back it. He has had to present to the emergency room twice recently when he was unable to push it back it. He was pending an outpatient repair of his right inguinal hernia.

What are the assessment and recommendations for this patient?

## 1 Introduction

Large language models (LLMs) (Brown et al., 2020; Chowdhery et al., 2022; Ouyang et al., 2022; OpenAI, 2022; 2023) have shown promise in the medical field (Singhal et al., 2022; Nori et al., 2023; Liévin et al., 2023). However, concerns about medical data privacy prevent the direct use of LLMs' medical capabilities in healthcare domain, as illustrated in Figure 1. Despite data usage policies[1] to safeguard user data, the implementation of these policies varies among LLMs, creating inconsistent protection levels. Moreover, medical data usage agreements[2] are stringent, explicitly forbidding data sharing with third parties, like direct uploads to ChatGPT (OpenAI, 2022). Therefore, developing methods to harness LLMs' medical knowledge while balancing data privacy in *privacy-restricted* scenarios is an urgent and underexplored research area.

---

\*Corresponding Author: xinluzhang@ucsb.edu
[1]https://openai.com/policies/api-data-usage-policies
[2]https://physionet.org/content/mimiciii/view-dua/1.4/

Figure 2: **Framework overview.** (a) To mitigate privacy leakage, we use a keyword extractor to obtain medical keywords. Clinicians then create several contexts based on these keywords and candidate answers, which the LLM uses to produce privacy-restricted contexts. (b) The generated contexts are used as additional input to enhance SLM medical decision-making capacity.

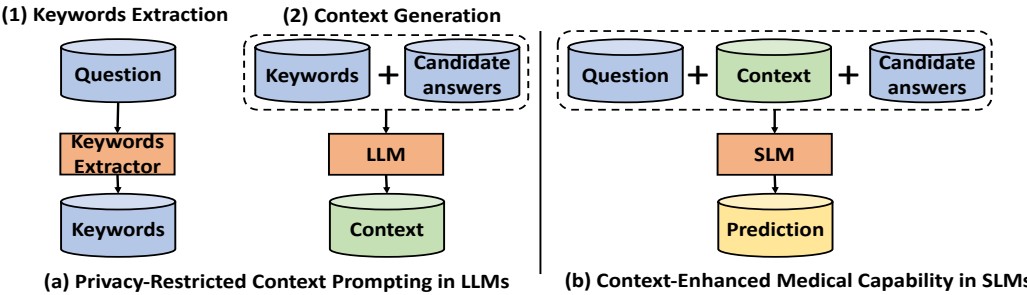

Small language models (SLMs) [3] that are specific to the medical domain (Gu et al., 2021; Yasunaga et al., 2022a; Lee et al., 2020; Alsentzer et al., 2019; Bolton et al., 2022; Lehman et al., 2023) have shown superior in-domain task performance compared to general-domain SLMs (Radford et al., 2019; Raffel et al., 2020; Devlin et al., 2019), addressing the vital need for data privacy in the medical field through local training. However, a notable performance gap between SLMs and LLMs in medical tasks remains (Singhal et al., 2022; Liévin et al., 2023). A critical question arises: How can we bridge the performance gap between SLMs and LLMs for medical tasks in privacy-restricted scenarios?

One common strategy to narrow the performance gap between LLMs and SLMs involves leveraging generated rationales (Wei et al., 2023; Wiegreffe et al., 2021) from LLMs to boost the performance of SLMs (Li et al., 2022; Fu et al., 2023; Ho et al., 2022; Shridhar et al., 2022). However, previous research has often required feeding complete data information into LLMs, ignoring privacy concerns. Thus, it is essential to explore alternative methods that can effectively utilize LLM-generated context, which is rich in medical knowledge, while balancing privacy concerns for LLMs and performance enhancement for SLMs in the medical domain.

In this paper, we present a simple yet effective pipeline that boosts SLM performance by incorporating medical contexts from LLMs in privacy-restricted scenarios. To the best of our knowledge, this is the first work that has utilized LLMs to improve SLM performance in such settings. While our primary focus is on multiple-choice medical QA, our framework can be adapted to other tasks or domains. Figure 2 illustrates our framework. Specifically, we use existing named-entity recognition (NER) models (Neumann et al., 2019) to extract keywords[4], thereby mitigating privacy risks. Based on these keywords and candidate answers, clinicians generate several medical contexts that mirror their thought processes. These clinician-written contexts serve as demonetisation to generate contexts for the remaining data by the LLM by in-context learning (Brown et al., 2020). Finally, we integrate these contexts into the SLMs to improve their performance on medical tasks.
Overall, our main contributions can be summarized as follows:

1. We propose a simple yet effective pipeline that uses keywords and candidate answers to elicit medical knowledge from LLMs, which is then fed into SLMs to enhance medical capabilities.
2. We introduce a privacy-conscious prompting strategy that mimics clinicians' thinking to generate medical knowledge-intensive contexts from LLMs in privacy-restricted scenarios.
3. Our method notably surpasses standard SLM fine-tuning without context in both full training and few-shot settings, achieving up to a 22.57% increase in accuracy, and obtains SOTA results in two medical tasks in privacy-restricted scenarios.

## 2 RELATED WORK

**LLMs in Biomedicine.** LLMs excel in NLP tasks, including medical fields (Brown et al., 2020; Chowdhery et al., 2022; Ouyang et al., 2022; OpenAI, 2022; 2023; Liévin et al., 2023; Kung et al., 2023; Singhal et al., 2022). Kung et al. (2023) used ChatGPT for the US Medical Exam. Liévin et al. (2023) leverage GPT-3.5 models for medical reasoning tasks. MedPaLM, tuned from FlanPaLM

---

[3]Following Li et al. (2022), we argue that the definition of small and large models is context-dependent.

[4]Keywords can be extracted by other methods, e.g., a manually created dictionary based on domain expertise.

(Chung et al., 2022), answered consumer medical questions comparably to clinicians (Singhal et al., 2022). However, privacy concerns limit LLMs in real-world medical uses, highlighting the need to utilize LLM medical knowledge in privacy-restricted setting.

**Biomedical SLMs.** Domain-specific SLMs, either extended from general-domain pretraining (Alsentzer et al., 2019; Lee et al., 2020; Gururangan et al., 2020) or built from scratch on biomedical corpora (Gu et al., 2021; Yasunaga et al., 2022a; Bolton et al., 2022), surpass general models (Radford et al., 2019; Raffel et al., 2020; Devlin et al., 2019) in biomedical tasks (Gu et al., 2021; Yasunaga et al., 2022a; Lee et al., 2020; Alsentzer et al., 2019; Bolton et al., 2022; Lehman et al., 2023). They also enhance privacy through local training. However, a performance gap remains between domain-specific SLMs and LLMs in medical tasks (Liévin et al., 2023; Singhal et al., 2022), emphasizing the need for strategies to reduce this gap.

**Knowledge Distillation from LLMs.** Previous studies have investigated distilling knowledge (Hinton et al., 2015) from LLMs to enhance smaller models' performance (Ho et al., 2022; Shridhar et al., 2022; Wang et al., 2022; Li et al., 2022). Ho et al. (2022) fine-tuned smaller models using InstructGPT-generated reasoning samples. Wang et al. (2022) input LLM generated rationales into SLM for question-answering. Li et al. (2022) applied LLM-derived explanations for multi-task learning. Zhang et al. (2023) tunes models for following medical instructions on data generated by GPT-4 and ChatGPT to better align with user intents across diverse medical applications. However, these approaches involve prompting LLMs without considering privacy preservation, which is vital in real-world medical scenarios (Bardhan et al., 2022; Shivade et al., 2019; Miura et al., 2020). We introduce a keyword-based prompting strategy to generate medical knowledge from LLMs, while balancing LLM privacy preservation and SLM performance enhancement.

**Augmenting NLP Tasks with External Knowledge.** Improving knowledge-intensive NLP tasks can be achieved by retrieving information from large evidence corpora like Wikipedia (Yu et al., 2023; Izacard and Grave, 2020; Yasunaga et al., 2022b;c; Guu et al., 2020; Lee et al., 2019a; Lewis et al., 2020). The retrieve-then-read model (Chen et al., 2017) utilizes retrievers, such as BM25 (Robertson et al., 1994) and DPR (Karpukhin et al., 2020), to identify relevant documents within a corpus. Then, a reader, like FiD (Izacard and Grave, 2020), analyzes these documents to enhance NLP tasks (Lewis et al., 2020; Qu et al., 2021; Sachan et al., 2021; Lee et al., 2019b). Some studies retrieve subgraphs from knowledge graphs to boost question-answering tasks (Yasunaga et al., 2022b; Zhang et al., 2022). Recent research shows that pre-trained language models can "retrieve" information via direct text generation (Petroni et al., 2019; Roberts et al., 2020). Liu et al. (2021); Yu et al. (2023) utilized LLMs to generate relevant contexts or background documents for question-answering tasks. Our work leverages LLMs as a knowledge base for medical knowledge retrieval.

**Data Privacy in BioNLP.** Biomedical data inherently contains sensitive information (Sousa and Kern, 2023). De-identification methods eliminate private details and replace them with synthetic data (Act, 1996; Chevrier et al., 2019; Leevy et al., 2020). For example, Liu et al. (2017); Dernoncourt et al. (2017) treat de-identification as a NER problem, modeling by neural networks. Despite being de-identified, sharing restrictions still apply (Johnson et al., 2016; 2020). Differential privacy (DP) provides theoretical bounds on individual data privacy while allowing aggregate statistical disclosure for the entire database (Dwork, 2008; Fernandes et al., 2019; Abadi et al., 2016). Melamud and Shivade (2019) train a model using a DP-based approach to generate synthetic clinical notes for secure sharing. In this work, we migrate privacy concerns during LLM inference by prompting the LLM with medical keywords extracted from raw data.

## 3 METHODOLOGY

### 3.1 PRIVACY-RESTRICTED CONTEXT PROMPTING IN LLMS

We consider a dataset $D = \{(q_i, A_i, y_i)\}^N$, where $N$ denotes the total number of instances, $q_i$ a problem, $A_i$ a set of candidate answers, and $y_i$ the correct answer. A set of keywords $k_i$ is extracted from each problem $q_i$ using a medical NER model (Neumann et al., 2019). To mitigate privacy leakage, we feed $k_i$ and $A_i$ into the LLM to generate medical context (Ho et al., 2022; Li et al., 2022).

We have a small set of clinician-written instances $E = \{(k_i^p, A_i^p, C_i^p, d_i^p)\}^M$, where both $C_i^p$, denoting medical contexts, and $d_i^p$, representing preliminary decisions, are generated based on partial data information $k_i^p$ and $A_i^p$, with $M \ll N$[5]. The medical context $C_i^p$ consists of an overall

---

[5]We set $M = 5$ for our experiments.

Figure 3: **LLM generates privacy-restricted medical contexts to enhance SLM decision-making.**
(a) The LLM generates medical knowledge-intensive context for each instance using clinicians'
few-shot demonstrations, extracted keywords from raw data ($k$), and candidate answers ($A$). The
generation output comprises: overall context ($c_o$), specific context of each candidate answer ($c_{a_j}$);
and preliminary decision of LLM ($d$). (b) The overall and specific contexts are then concatenated ($\oplus$)
with the question as additional input to fine-tune a SLM, enhancing its medical decision-making.

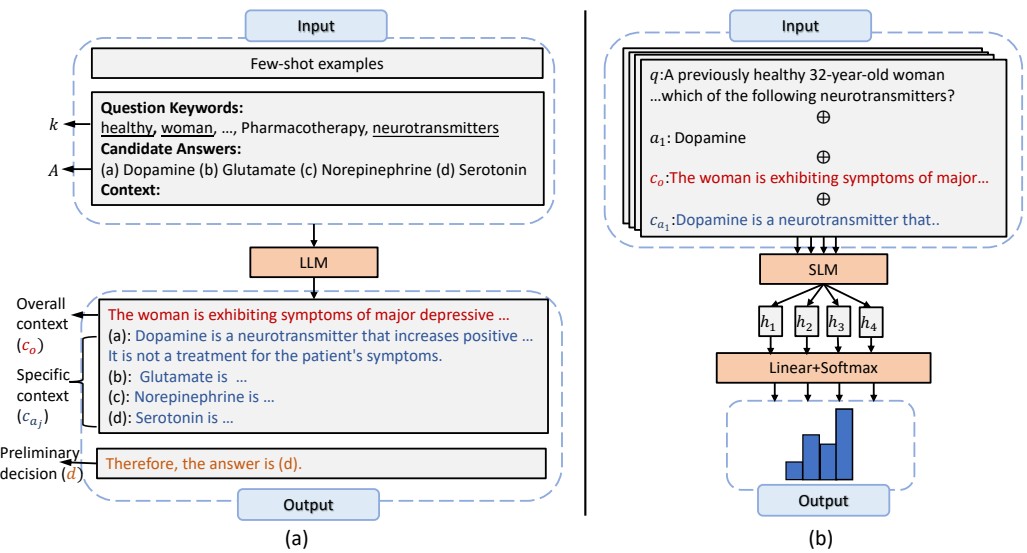

(a)                                         (b)

context and specific contexts for each candidate answer. The overall context encapsulates high-level
medical knowledge based on keywords and candidate answers, while specific contexts provide
detailed information for each candidate answer and its relationship to the overall context. The
preliminary decision represents clinicians' predictions informed by these contexts. This prompt
strategy simulates clinicians' reasoning steps by producing high-level guidance based on partial data,
examining individual candidate answers in-depth, and ultimately making an initial determination
given prior analysis, as shown in Figure 3 (a).[6].

We utilize the LLM with $E$ as demonstrations for in-context learning to generate medical contexts
and preliminary decisions for all instances in $D$. For $1 \leq i \leq N$, we concatenate all instances in
$E$, $k_i$, and $A_i$ and then feed this concatenated string into the LLM for decoding. After this, we
parse the decoded sentence into two parts: the context part $C_i$, which includes overall and specific
contexts, and the preliminary decision part $d_i$[7], which serves as a performance metric for the LLM.
All instances of $C_i$ are used for SLM training, regardless of the LLM's preliminary decision accuracy.
We find that these contexts contain valuable medical knowledge even with incorrect decisions. We
defer more details of our experiments into Section 5.1.

### 3.2 CONTEXT-ENHANCED MEDICAL CAPABILITY IN SLMS

Given an augmented dataset $D' = \{(q_i, A_i, C_i, y_i)\}^N$, our goal is to leverage the medical context $C_i$
generated by the LLM to enhance the SLM's medical proficiency and predict $y_i$ for each instance.
We omit $i$ for simplicity. Inspired by previous works (Wang et al., 2022), we treat $C$ as additional
input for the SLM to aid the decision-making. For each candidate answer $a_j \in A$, $0 \leq j < |A|$,
we provide both the overall context, $c_o$, and the specific context of the answer, $c_{a_j}$, concatenating
these contexts with the question $q$ and answer $a_j$. The SLM generates a contextual representation
vector $\mathbf{h}_j$ for each choice, which is then fed into a linear layer to produce $s_j$, a prediction score for
the correctness of the answer choice:

$$\mathbf{h}_j = \text{SLM}([q \oplus a_j \oplus c_o \oplus c_{a_j}]), \; s_j = \text{Linear}(\mathbf{h}_j).$$

For each $a_j$, the score $s_j$ is computed and normalized using the softmax function across all candidate
answers, as shown in , as shown in Figure 3 (b). During training, models are optimized to maximize
correct answer scores employing standard cross-entropy loss between predictions and ground truths.

---

[6]See Appendix A.7 for prompt details.

[7]We only use $d_i$ to evaluate LLM performance and do not add this into SLM training.

In the inference phase, $s_j$ is calculated for each $a_j$, and the answer with the highest score is the predicted answer.

## 4 EXPERIMENTS

### 4.1 EXPERIMENTAL SETUP

**Datasets.** We evaluate our methods on the first three datasets for in-domain performance and on all four datasets for out-of-domain performance: **1. MedQA** (Jin et al., 2020) contains 4-way multiple-choice questions from the US Medical Licensing Exam. It has 10,178/1,272/1,273 instances in the training/development/test sets. Results on the development and test sets are reported. **2. HEADQA** (Vilares and Gómez-Rodríguez, 2019) features multiple-choice questions from specialized Spanish healthcare exams conducted between 2013 and 2017. The dataset has 2,657/1,366/2,742 instances in the training/development/test sets. We report results on the development and test sets. **3. MedMCQA** (Pal et al., 2022) is a 4-way multiple-choice dataset from Indian medical school entrance exams. It has 182.8k/4.2k/6.1k instances in the training/development/test sets. We use a randomly selected subset of 10,000 training instances and report results on the development set, following previous work (Nori et al., 2023). **4. MMLU-professional medicine** (Hendrycks et al., 2021) is a 4-way biomedical multiple-choice dataset, with 5/31/272 instances in the training/validation/test sets. We evaluate the Out-of-Domain (OOD) performance of our method on the test set without adaptation.

**Context Generation with LLM.** We use the *gpt-3.5-turbo* via OpenAI API[8] and employ greedy decoding for in-context learning. Each dataset has five-shot medical examples, shown in Figure 3.

**Training SLMs with Contextual Information.** After acquiring data from LLMs, we utilize BioLinkBert-Base (Yasunaga et al., 2022a), BioLinkBert-Large (Yasunaga et al., 2022a), and BioMedLM (Bolton et al., 2022) as SLM backbones for Fine-Tuning with Context (FTC). We compare FTC with privacy-restricted baselines that leverage additional knowledge to aid medical decision making: QA-GNN (Yasunaga et al., 2022b), GreaseLM (Zhang et al., 2022), DRAGON (Yasunaga et al., 2022c), MurKe (Miura et al., 2020), MOEBQA (Dai et al., 2022), HDRN (Mao et al., 2022) and VOD (Liévin et al., 2022). Also, we perform SLM standard fine-tuning (SFT) without any external knowledge and LLM prompting with keywords and candidate answers (LLM) to validate our method's efficacy. To ensure a fair comparison, we keep the backbones and hyperparameters consistent for both FTC and SFT approaches. For BioLinkBERT-Base, We conduct three separate runs for each setting and report the average results along with the standard deviation. We report only a single run for BioLinkBERT-Large and BioMedLM due to high computational cost.[9]. The performance are measured by accuracy (%).

### 4.2 SUPERIOR MEDICAL DECISION PERFORMANCE OF FTC

Table 1 compare results between Fine-Tuning with Context (FTC) and baselines. FTC significantly outperforms standard fine-tuning (SFT), with improvements of up to 7.96%, 21.23%, and 17.05% in absolute accuracy on the test sets of MedQA and HEADQA, and the development set of MedMCQA, respectively. Furthermore, FTC exceeds LLM by 14.20%, 15.75%, and 16.90% on these datasets in privacy-restricted scenarios. These results indicate that SLMs effectively leverage the medical knowledge provided by the LLM to aid decision-making, highlighting the benefits of incorporating context from the LLM in SLMs training process.

FTC with BioMedLM achieves SOTA performance on both MedQA and HEADQA datasets. Particularly, for HEADQA, FTC with BioMedLM backbone outperforms the best baseline MOEBQA (Dai et al., 2022) by 18.80% and 16.55% in absolute accuracy on the development and test sets, respectively. This demonstrates the considerable impact of the FTC on enhancing the performance of SLMs in medical tasks. Compared to the complex VOD Liévin et al. (2022) baseline, a retriever-and-reader framework with multiple training strategies, our approach is more straightforward. We simply fine-tune SLMs and include only one context per candidate answer generated by the LLM. Despite this, our method achieves superior performance in MedQA and secures second place in MedMCQA, utilizing less than 6% of the training data required by the VOD [10].

---

[8]https://platform.openai.com/docs/models/gpt-3-5

[9]We provide implementation and training details in Appendix A.2.

[10]In theory, FTC could be integrated with VOD Liévin et al. (2022). We intend to do this once their code is ready.

Table 1: Performance comparison (%) on MedQA, HEADQA and MedMCQA. Best results are bold and second best are underline. †: results from their original papers. ¶: results from Yasunaga et al. (2022c). ‡ results from Bolton et al. (2022). §: VOD uses 180k training instances for MedMCQA, in contrast to our approach which utilizes only 10k instances.

| | MedQA | | HEADQA | | MedMCQA |
|---|---|---|---|---|---|
| | dev | test | dev | test | |
| QA-GNN (Yasunaga et al., 2022b) | - | 45.0¶ | - | - | - |
| GREASELM (Zhang et al., 2022) | - | 45.1¶ | - | - | - |
| DRAGON (Yasunaga et al., 2022c) | - | 47.5¶ | - | - | - |
| HDRN(Mao et al., 2022) | - | 47.6† | - | - | - |
| MurKe (Liu et al., 2020) | - | - | - | 46.7† | - |
| MOEBQA (Dai et al., 2022) | 39.9† | 41.6† | 44.3† | 46.7† | - |
| VOD (Liévin et al., 2022) | | | | | |
| + BioLinkBERT&BM25 | 41.0† | 40.4† | - | - | 51.6† § |
| + BioLinkBERT& BioLinkBERT | $\underline{53.6}$† | 55.0† | - | - | **58.3**† § |
| LLM | 38.30 | 41.70 | 47.60 | 47.50 | 35.20 |
| SFT (w/o addtional knowledge) | | | | | |
| + BioLinkBERT-Base | $41.22_{0.48}$ | $42.21_{0.91}$ | $39.14_{1.88}$ | $41.00_{0.34}$ | $32.15_{2.23}$ |
| + BioLinkBERT-Large | - | 45.1‡ | 39.53 | 41.61 | 35.86 |
| + BioMedLM | - | 50.3‡ | 48.68 | 50.33 | 43.63 |
| FTC (Ours) | | | | | |
| + BioLinkBERT-Base | $50.73_{0.35}$ | $50.17_{0.42}$ | $61.35_{0.16}$ | $60.21_{0.47}$ | $49.20_{0.45}$ |
| + BioLinkBERT-Large | 51.02 | 53.10 | $\underline{62.30}$ | $\underline{62.18}$ | 50.38 |
| + BioMedLM | **53.85** | **55.90** | **63.10** | **63.17** | $\underline{52.09}$ |

Table 2: Results (%) of LLM, SFT and FTC under different training sizes. [11]

| | MedQA | | | | HEADQA | | | | MedMCQA | | | |
|---|---|---|---|---|---|---|---|---|---|---|---|---|
| | 100 | 200 | 500 | full | 100 | 200 | 500 | full | 100 | 200 | 500 | full |
| LLM | | | 41.70 | | | | 47.50 | | | | 35.20 | |
| SFT | $31.40_{0.79}$ | $33.79_{1.34}$ | $35.27_{0.63}$ | $42.21_{0.91}$ | $36.54_{0.30}$ | $39.00_{0.61}$ | $34.88_{2.00}$ | $41.00_{0.34}$ | $29.07_{0.16}$ | $28.31_{0.86}$ | $31.43_{1.32}$ | $32.15_{2.23}$ |
| FTC | **$45.52_{1.03}$** | **$46.29_{0.32}$** | **$47.87_{1.41}$** | $50.17_{0.42}$ | **$55.03_{1.10}$** | **$56.18_{0.76}$** | **$57.45_{0.53}$** | $60.21_{1.47}$ | **$38.82_{1.03}$** | **$40.12_{0.58}$** | **$43.06_{1.92}$** | $49.20_{0.45}$ |

## 4.3 FEW-SHOT LEARNING ENHANCEMENT WITH FTC

Real medical environments often face scarce training data (Zhang et al., 2021; Ge et al., 2022). In this section, we explore if additional contexts boost SLM's medical proficiency in few-shot setting. We experiment on BioLinkBERT-Base with training sample sizes of $\{100, 200, 500\}$ for all datasets. For every size, we randomly generate three data splits from the entire training set, performing a single run for each split. Results are shown in Table 2.

FTC consistently surpasses SFT by a considerable margin, achieving absolute accuracy enhancements of up to 14.12%, 22.57%, and 11.81% for the test sets of MedQA and HEADQA, and the development set of MedMCQA, respectively. These consistent gains demonstrate that our method not only enhances performance in full-training but also proves highly beneficial when training data is limited. Interestingly, SFT with full training data does not surpass LLM in HEADQA and MedMCQA, and achieves only a comparable performance to LLM in MedQA.

In contrast, our FTC method consistently surpasses LLM and SFT with full training data in all tasks, even with as few as 100 training data points. This underscores the efficacy of prompting medical knowledge from the LLM to boost the SLM's medical capacities.

## 4.4 OUT-OF-DOMAIN (OOD) GENERALIZABILITY BOOST WITH FTC

To evaluate the generalizability of our approach, we investigate the OOD performance of FTC using BioLinkBERT-Base as the backbone, without additional training. The best model from the source domain in Section 4.2 is directly applied to the target domain. Table 3 presents the OOD performance.

---

[11]We present results for the test sets of all datasets, excluding MedMCQA. For those datasets with available development set results, the results are provided in the Appendix A.3.

Table 3: Accuracy comparison (%) between LLM on the target domain (upper), and FTC and SFT trained on a source domain (lower) and applied directly to the target domain.

| | MedQA | | HEADQA | | MedMCQA | | MMLU | | |
| | HEADQA | MedMCQA | MedQA | MedMCQA | MedQA | HEADQA | MedQA | HEADQA | MedMCQA |
|---|---|---|---|---|---|---|---|---|---|
| LLM | 41.70 | | 47.50 | | 35.20 | | 52.94 | | |
| SFT | $35.57_{0.24}$ | $31.32_{2.38}$ | $35.62_{3.19}$ | $34.34_{4.07}$ | $30.14_{1.07}$ | $31.77_{0.37}$ | $41.30_{6.89}$ | $38.84_{2.13}$ | $32.97_{6.07}$ |
| FTC | $\mathbf{47.26}_{0.96}$ | $\mathbf{49.25}_{0.17}$ | $\mathbf{55.27}_{1.28}$ | $\mathbf{61.90}_{0.38}$ | $\mathbf{41.14}_{0.26}$ | $\mathbf{45.98}_{0.78}$ | $\mathbf{58.95}_{1.73}$ | $\mathbf{54.53}_{2.11}$ | $\mathbf{54.66}_{0.17}$ |

The OOD performance of SFT is inferior compared to LLM prompting in the target domain, indicating its limited generalization capabilities. Conversely, FTC consistently outperforms both SFT in OOD settings and LLM prompting baselines. This underscores the enhanced generalizability achieved by incorporating the medical context generated by the LLM into the SLM.

## 4.5 Strong General Applicability of FTC

Privacy concerns not only appear in the medical domain. In this section, we investigate whether our method is generally applicable beyond the medical domain. We use two general domain datasets, CommonsenseQA (CSQA) (Talmor et al., 2018) and OpenbookQA (OBQA) (Mihaylov et al., 2018), under privacy-restricted settings in the full training scenarios. We adopt T5-base (Raffel et al., 2020) as the SLM backbone following previous works (Li et al., 2022; Wang et al., 2022) and utilize Fusion-in-Decoder (Izacard and Grave, 2020) to incorporate contexts. We conduct three separate runs for each setting[12]. Table 4 shows the results.

Table 4: Accuracy comparison (%) in general domain.

| | CSQA | OBQA |
|---|---|---|
| LLM | 41.25 | 51.60 |
| SFT | $62.63_{0.17}$ | $56.93_{0.25}$ |
| FTC | $65.87_{0.23}$ | $68.60_{1.43}$ |

FTC consistently outperforms LLM and SFT baselines on two datasets. Specifically, FTC performs 3.24% and 11.67% better than its standard finetuning counterpart, SFT, in CSQA and OBQA, respectively. This demonstrates the effectiveness of our method, utilizing the LLM as a strong knowledge base and prompting knowledge within the LLM in privacy-restricted scenarios, which in turn enhances the SLM's knowledge capacities and improves decision-making.

## 5 Analysis

### 5.1 Context Analysis

To further understand the effectiveness of context generated by the LLM on SLM performance, we perform ablation studies with BioLinkBert-Base as the SLM backbone.

**Role of Context Parts.** We investigate the effectiveness of each part of the context by separately providing (1) overall context (*Only Overall*) and (2) specific context for each candidate answer (*Only Specific*) as additional information to the SLM in both few-shot and full-training settings. Results are shown in the upper part (a) of Figure 4. Despite decreased performance when providing only overall or specific context compared to FTC, SLMs with added medical context still outperform SFT baselines across various training settings, demonstrating the importance of both context aspects in informed decision-making. The more pronounced performance drop for overall context in most settings could be attributed to it offering general medical knowledge, while specific context provides tailored knowledge for each candidate answer.

**Content Learned by SLM.** Specific context for each candidate answer includes its relationship to overall context, as shown in Figure 3. For example, the relation of answer (a) is "It is not a treatment for the patient's symptoms." We explore whether the SLM learns from medical knowledge or merely cherry-picks relationships by removing all relationship information in specific contexts (*No Relation*), retaining only the knowledge content. Then, we train the SLM using these modified contexts in both few-shot and full-training scenarios. Results are shown in the lower part (b) of Figure 4. When relationships are removed, performance declines compared to the FTC. However, even without any relationship information, the SLM with medical contexts consistently and significantly outperforms the SFT. This suggests that, although relationships assist in decision-making, the SLM prioritizes medical knowledge from context over simply replicating relationships directly.

**Context Case Study.** To examine what medical knowledge is generated by the LLM using keywords and candidate answers, and how FTC reasons with these generated contexts, we analyzed instances

---

[12]We defer detailed experiment settings to the Appendix A.6.

Figure 4: Results of ablation studies [13]. The upper part examines the effect of context components on SLM training, while the lower part investigates the impact of relationships within the context.

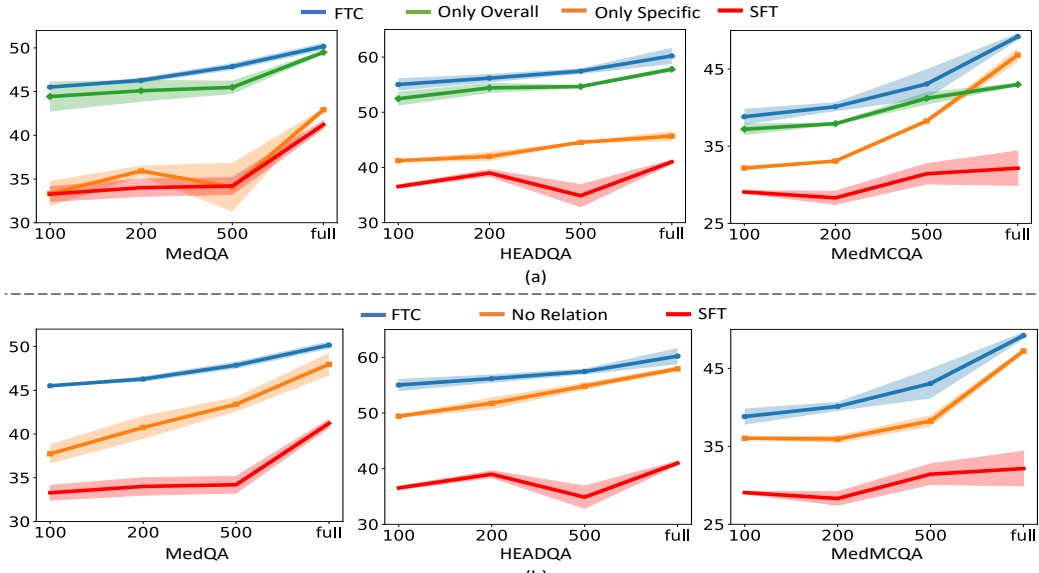

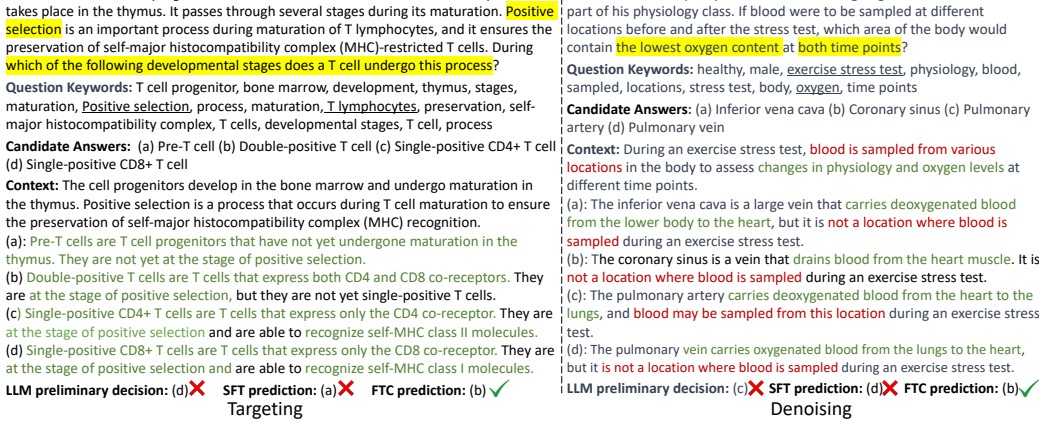

Figure 5: Case Study: MedQA test set contexts and predictions. yellow highlights important local information; underlined indicates LLM-selected keywords for context generation; green and red signify correct and incorrect contexts that could aid or confuse the SLM, respectively. FTC succeeded 113 instances where SFT and LLM failed: 45 *Targeting* (left) and 68 *Denoising* (right).

**Question:** After a T cell progenitor leaves the bone marrow, most of its development takes place in the thymus. It passes through several stages during its maturation. Positive selection is an important process during maturation of T lymphocytes, and it ensures the preservation of self-major histocompatibility complex (MHC)-restricted T cells. During which of the following developmental stages does a T cell undergo this process?

**Question Keywords:** T cell progenitor, bone marrow, development, thymus, stages, maturation, Positive selection, process, maturation, T lymphocytes, preservation, self-major histocompatibility complex, T cells, developmental stages, T cell, process

**Candidate Answers:** (a) Pre-T cell (b) Double-positive T cell (c) Single-positive CD4+ T cell (d) Single-positive CD8+ T cell

**Context:** The cell progenitors develop in the bone marrow and undergo maturation in the thymus. Positive selection is a process that occurs during T cell maturation to ensure the preservation of self-major histocompatibility complex (MHC) recognition.
(a): Pre-T cells are T cell progenitors that have not yet undergone maturation in the thymus. They are not yet at the stage of positive selection.
(b) Double-positive T cells are T cells that express both CD4 and CD8 co-receptors. They are at the stage of positive selection, but they are not yet single-positive T cells.
(c) Single-positive CD4+ T cells are T cells that express only the CD4 co-receptor. They are at the stage of positive selection and are able to recognize self-MHC class II molecules.
(d) Single-positive CD8+ T cells are T cells that express only the CD8 co-receptor. They are at the stage of positive selection and are able to recognize self-MHC class I molecules.

**LLM preliminary decision:** (d) ✘     **SFT prediction:** (a) ✘     **FTC prediction:** (b) ✔
Targeting

**Question:** A healthy 23-year-old male is undergoing an exercise stress test as part of his physiology class. If blood were to be sampled at different locations before and after the stress test, which area of the body would contain the lowest oxygen content at both time points?

**Question Keywords:** healthy, male, exercise stress test, physiology, blood, sampled, locations, stress test, body, oxygen, time points

**Candidate Answers:** (a) Inferior vena cava (b) Coronary sinus (c) Pulmonary artery (d) Pulmonary vein

**Context:** During an exercise stress test, blood is sampled from various locations in the body to assess changes in physiology and oxygen levels at different time points.
(a): The inferior vena cava is a large vein that carries deoxygenated blood from the lower body to the heart, but it is not a location where blood is sampled during an exercise stress test.
(b): The coronary sinus is a vein that drains blood from the heart muscle. It is not a location where blood is sampled during an exercise stress test.
(c): The pulmonary artery carries deoxygenated blood from the heart to the lungs, and blood may be sampled from this location during an exercise stress test.
(d): The pulmonary vein carries oxygenated blood from the lungs to the heart, but it is not a location where blood is sampled during an exercise stress test.

**LLM preliminary decision:** (c) ✘     **SFT prediction:** (d) ✘     **FTC prediction:** (b) ✔
Denoising

from the MedQA test set where the FTC correctly predicted answers while both the SFT and LLM failed. Clinicians identified two distinct categories among the cases: (1) *Targeting*, where the LLM successfully refines the target scope and generates high-quality medical contexts, albeit arriving at an incorrect answer; the SLM integrates these contexts with the raw question and correctly predicts the answer. (2) *Denoising*, where the LLM fails to figure out the correct relationship of the correct answer and generates noisy medical knowledge; the SLM effectively obtains useful information, combines it with localized data, and ultimately makes the correct prediction. Figure 5 provides examples of each case. This case study demonstrates that LLMs can generate valuable medical information even when making incorrect decisions based on partial data, and that the FTC can extract useful medical knowledge from noisy contexts, thereby enhancing SLM medical reasoning capabilities.

**Context Quality.** To further quantitatively demonstrate that LLM-generated context retains medical knowledge even with incorrect preliminary decisions. We compared FTC and *Fine-Tuning with Context and Rejection* (FTCR) in full training settings. FTCR uses context for SLM training only if the LLM's preliminary decision is correct; otherwise, no additional context is provided. The results are in Table 5. FTC consistently outperforms FTCR across three medical tasks, implying valuable

Table 5: Performance comparison of FTC and FTCR in the full-training setting [14].

|  | MedQA | HEADQA | MedMCQA |
|---|---|---|---|
| FTCR | $48.12_{0.66}$ | $57.79_{0.65}$ | $46.55_{0.28}$ |
| FTC | $\mathbf{50.17}_{0.42}$ | $\mathbf{60.21}_{1.47}$ | $\mathbf{49.20}_{0.45}$ |

Table 6: Privacy budget statistics in MedQA. Avg. K and Avg. Q are the average word count for keywords and raw questions, respectively, across the dataset. Budget is privacy budget.

|  | Avg. K | Avg. Q | Budget |
|---|---|---|---|
| Train + Dev | 49.1 | 116.2 | 42.3% |
| Test | 50.7 | 119.6 | 42.4% |

Table 7: Results of different information representation methods, maintaining the same privacy budget on MedQA.

|  | Dev | Test |
|---|---|---|
| LLM prompting |  |  |
| + Random Span | 27.59 | 28.52 |
| + Random Words | 30.03 | 30.48 |
| + Keywords | 38.30 | 41.70 |
| SLM fine-tuning |  |  |
| SFT | $35.67_{0.47}$ | $33.99_{0.87}$ |
| FTC |  |  |
| + Random Span | $42.95_{0.33}$ | $44.10_{0.80}$ |
| + Random Words | $44.18_{1.11}$ | $45.06_{1.38}$ |
| + Keywords | $46.42_{0.28}$ | $47.91_{0.51}$ |

Figure 6: LLM and FTC results under different keyword usage ratios on MedQA. Standard divisions of FTC and SFT are omitted for simplicity.

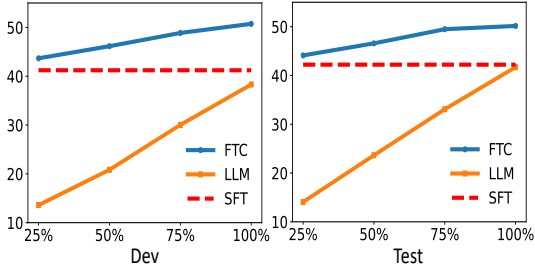

medical knowledge remains in contexts even with LLM's incorrect decisions. SLM can harness these insights from impfect contexts to enhance medical capabilities.

## 5.2 PRIVACY ANALYSIS

We conduct a privacy analysis on MedQA using BioLinkBERT-Base and introduce the *privacy budget*, a metric estimating information usage, presented in Table 6. The privacy budget is calculated as the ratio of the number of words provided to the LLM to the total words in the original question. Lower privacy budgets signify better privacy preservation [15].

**Why Keywords?** We evaluate the effectiveness of using keywords (*Keywords*) to represent raw data when querying LLMs and compare it to two other methods of raw data representation: (1) random consecutive word spans (*Random Span*), and (2) random word bags from the original data (*Random Words*). Given the same privacy budget, we randomly select a shared set of 1000 training instances for each method, use these to query the LLM for medical contexts, and then use these contexts as additional input for SLM training. Table 7 displays the accuracy of LLM prompting and SLM fine-tuning. All FTC methods consistently outperform the SFT baselines, demonstrating their effectiveness. Keywords perform better than Random Span and Random Words, providing a more efficient representation of medical knowledge within the same privacy budget. LLM prompting performance parallels SLM training, underscoring the importance of raw data representation in context generation and effective SLM training.

**Privacy Budget-Model Performance Trade-off.** We analyze the trade-off between privacy budget and model performance by generating context from the LLM using randomly selected {25%, 50%, 75%, 100%} of keywords and training the SLM with full training data and corresponding context. Figure 6 displays the accuracy for LLM prompting and SLM fine-tuning at various privacy budgets. As privacy budget increases, performance improves. Impressively, FTC outperforms SFT using context from just 25% of keywords, resulting in LLM prompting performance below 15%—significantly lower than the 25% random guess rate. This suggests that LLM-generated context maintains essential medical knowledge despite limited raw data information, and the SLM effectively learns from it.

## 6 CONCLUSION

We introduce a simple yet effective pipeline that enhances the SLM performance in medical tasks by using medical keywords to prompt LLMs within privacy-restricted scenarios. Our experimental results across three medical tasks in various training settings underscore the effectiveness of our proposed approach. Through a comprehensive analysis, we gain a deeper understanding of our method capabilities and the impact of LLMs on SLM performance in privacy-restricted scenarios.

---

[14]We present results for the test sets of all datasets, excluding MedMCQA. For those datasets with available development set results, the results are provided in the Appendix A.3.

[15]We defer BPC evaluation result on privacy to appendix A.4

ACKNOWLEDGMENTS

We would like to express our gratitude to Zhiyu Chen from Meta Reality Labs, Ming Yi, and Hong Wang from the Computer Science Department at UCSB, as well as the anonymous reviewers, for their invaluable feedback. Additionally, we extend our thanks to Rachael A Callcut and Anamaria J Roble for their insightful discussions and guidance on medical prompt designs. Furthermore, we gratefully acknowledge the generous financial support provided by the National Institutes for Health (NIH) grant NIH 7R01HL149670.

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

LIMITATIONS

While our work enhances SLM performance by using keyword representations of raw data, it only mitigates but does not eliminate privacy concerns. Given that FTC is based on GPT3.5, the medical knowledge it generates may be inaccurate or biased, which can impact SLM performance. Moreover, the inference time for LLMs may be slower than that of SLMs, leading to longer overall inference times compared to models solely reliant on local SLMs. Furthermore, due to the training cut-off time, the medical knowledge in LLM could be outdated, potentially hindering medical decision-making. We aim to integrate LLM with the internet and knowledge graph in future work to generate more reliable medical knowledge for enhancing SLM decision-making. These issues underscore the need for further research on the use of LLMs in privacy-restricted medical scenarios.

## A APPENDIX

### A.1 DATA AND CODE

Our codes and generated data are public at `https://github.com/XZhang97666/PrivacyBoost-SLM`.

### A.2 SLM IMPLEMENTATION AND TRAINING DETAILS

We implement both SFT and FTC based on huggingface transformers Wolf et al. (2020), and train on NVIDIA A40-48GB GPUs. For all datasets, we utilize AdamW (Loshchilov and Hutter, 2019) as optimizer. For MedQA and HEADQA, we set learning rates of $5 \times 10^{-5}$, $5 \times 10^{-5}$, and $2 \times 10^{-6}$ for BioLinkBERT-Base, BioLinkBERT-Large, and BioMedLM in both FTC and SFT settings. For MedMCQA, we set learning rates of $2 \times 10^{-5}$, $2 \times 10^{-5}$, and $2 \times 10^{-6}$ for BioLinkBERT-Base, BioLinkBERT-Large, and BioMedLM in both FTC and SFT settings. For BioLinkBERT-Base and BioLinkBERT-Large, we limit training to 100 epochs with a 200-step warm-up and apply early stopping after 5 epochs without validation improvement. Batch sizes are 8 for few-shot and full-training scenarios across all datasets. For BioMedLM, we set the training epochs to 10 for all datasets. We run experiments with three random seeds {0, 1, 2} and report mean results and standard deviations.

### A.3 ADDITIONAL EXPERIMENTAL RESULTS

**Development sets results of MedQA and HeadQA.** We report development sets results of MedQA and HeadQA in Table 8 and 9, and Figure 7.

Table 8: Results (%) on development sets of MedQA and HEADQA between LLM, SFT, and FTC under different training sizes.

| | MedQA | | | | HEADQA | | | |
|---|---|---|---|---|---|---|---|---|
| | 100 | 200 | 500 | full | 100 | 200 | 500 | full |
| LLM | | 38.30 | | | | 47.60 | | |
| SFT | $33.28_{0.85}$ | $34.01_{1.00}$ | $34.20_{0.96}$ | $42.21_{0.91}$ | $36.54_{0.30}$ | $39.00_{0.61}$ | $34.88_{2.00}$ | $41.48_{0.48}$ |
| FTC | $\mathbf{43.66}_{1.26}$ | $\mathbf{45.70}_{0.10}$ | $\mathbf{45.62}_{0.70}$ | $\mathbf{50.73}_{0.35}$ | $\mathbf{55.03}_{1.10}$ | $\mathbf{56.18}_{0.76}$ | $\mathbf{57.45}_{0.53}$ | $\mathbf{60.21}_{1.47}$ |

Table 9: Results comparison of FTC and FTCR in the full-training setting on development sets of MedQA and HEADQA.

| | MedQA | HEADQA |
|---|---|---|
| FTCR | $48.56_{0.62}$ | $58.17_{0.78}$ |
| FTC | $\mathbf{50.73}_{0.35}$ | $\mathbf{61.35}_{0.16}$ |

Figure 7: Accuracy comparison (%) of ablation studies on development sets of MedQA and HEADQA. The upper part of the table examines the effect of different context components on SLM training, while the lower part investigates the impact of relationships within the context.

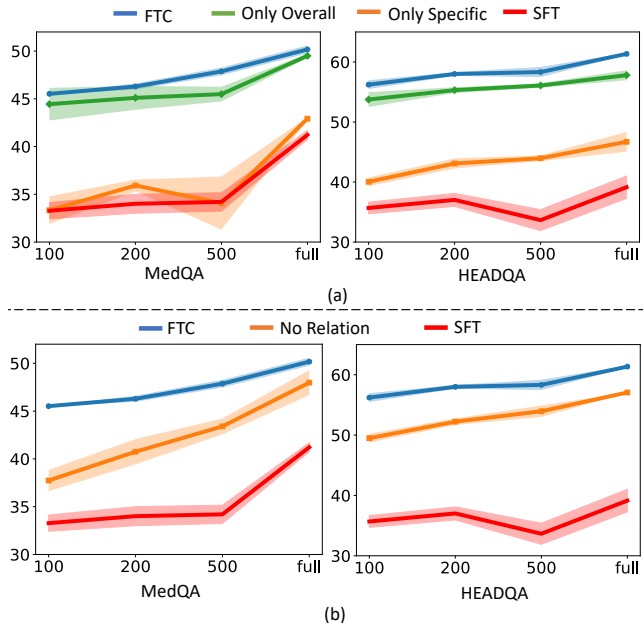

### A.4 ALTERNATIVE MEASUREMENT FOR PRIVACY BUDGET

We further conducted BPC measurement experiments on the training, validation, and test sets of the MedQA dataset, respectively, to demonstrate the effectiveness of using keywords to represent raw medical data while keeping privacy. To evaluate the BPC of raw data and keywords, we separately input keywords with various proportions and raw data into BioMedLM across different sets. Specifically, for each list of keywords, we form a sentence of keywords by concatenating the list of keywords and separating each pair of keywords with an empty space. Subsequently, we calculate the corresponding BPC values to assess the outcomes. The results are shown in Table 10.

Table 10: Comparison of BPC values for raw data and concatenated keywords across different data splits of the MedQA dataset

|  | Raw Data | Keywords | | | |
|  | - | 25% | 50% | 75% | 100% |
|---|---|---|---|---|---|
| Training | 12.41 | 10.37 | 10.04 | 9.91 | 9.87 |
| Validation | 12.44 | 10.31 | 10.01 | 9.92 | 9.89 |
| Test | 12.44 | 10.32 | 10.06 | 9.94 | 9.91 |

**The BPC values of keywords consistently exhibit lower values than those of the raw data.**

A higher BPC is consistently observed across various subsets of the MedQA dataset when compared to the approach of inputting keywords into BioMedLM. This comparison implies that, on average, raw data holds a greater level of uncertainty in contrast to the utilization of keywords. This disparity could be attributed to the fact that raw data encompasses a larger amount of medical-unrelated information, which includes privacy-related data, despite its comprehensive information coverage.

**A decrease in the proportion of keywords results in an increase in BPC.**

Decreasing the number of keywords fed into the BioMedLM leads to an increase in BPC. This indicates that a higher volume of keywords contributes to a more meaningful representation of medical information in raw data, subsequently enhancing the performance of both LLM and FTC.

**Keywords representation obtains the lowest BPC compared to random words and random span.**
We further evaluate the BPC of different raw data representation methods (random span, random words and keywords) while maintaining the privacy budget the same. Specifically, we feed different data representation content into BioMedLM on the training, validation and test sets of the MedQA dataset, and calculate the corresponding BPC values respectively. The results are presented in Table 11.

Table 11: BPC values different raw data representation methods.

| Representation Methods | Keywords | Random words | Random span |
|---|---|---|---|
| Training | 9.87 | 10.04 | 12.25 |
| Validation | 9.89 | 10.05 | 12.22 |
| Test | 9.91 | 10.04 | 12.22 |

Keyword representation consistently obtains the lowest BPC compared to the other two data representation methods, providing a more effective representation of medical knowledge within the same privacy budget. Interestingly, the order of BPC performance aligns with the performance of LLM prompting and SLM fine-tuning in Table 7 of our paper. The method with a lower BPC value achieves better performance in both LLM prompting and SLM fine-tuning. This further highlights the importance of raw data representation in generating high-quality context and effective SLM training.

### A.5 EXPERIMENTS ON WHAT SLM LEARNS FOR DECISION MAKING.

We feed preliminary decisions (PD) as context into SLM with backbone BioLinkBert-Base on three datasets. Three separate runs for each setting are conducted and the average results along with the standard deviation are reported. The results are shown in the Table 12.

Table 12: Results comparison of SLM with preliminary decisions and FTC.

| | MEDQA | HeadQA | MEDMCQA |
|---|---|---|---|
| SLM w PD | $47.21_{0.31}$ | $53.64_{1.09}$ | $45.42_{0.17}$ |
| FTC | $\mathbf{50.17}_{0.42}$ | $\mathbf{61.35}_{0.16}$ | $\mathbf{49.20}_{0.45}$ |

FTC, which integrates extensive medical knowledge into the decision-making, shows a consistent and significant improvement over the SLM that only uses PD for context. These findings underscore the valuable contribution of leveraging comprehensive medical knowledge, provided by LLM, in enhancing the medical decision-making capabilities.

### A.6 GENERAL DOMAIN EXPERIMENTAL SETUP

We perform experiments on two commonsense datasets to demonstrate the broad applicability of our approach.

**Datasets. 1. CommonsenseQA** (Talmor et al., 2018) is a multi-choice question-answering dataset featuring 5 options per question, requiring commonsense reasoning. The dataset is split into 9741/1221/1140 instances for training, development, and test sets, respectively. As the test set is not publicly accessible, we follow previous work (Li et al., 2022) and report results on the development set. **2. OpenbookQA** is a 4-option multi-choice question-answering dataset that demands open book facts, broad common knowledge, and multi-hop reasoning (Mihaylov et al., 2018). The dataset is split into 4957/500/500 instances for training, development, and test sets, respectively. We report results on the test set.

**Context generation from LLM.** We utilize the grounded entities from Zhang et al. (2022) as keywords to query the GPT-3.5 *gpt-3.5-turbo* engine, generating context through a greedy decoding process (by setting the temperature to 0). We employ the same demonstration format as in the medical domain and create a privacy-restricted context in accordance with the in-context learning paradigm. For each dataset, we supply seven-shot hand-crafted examples.

**SLM training.** We employ T5-base (Raffel et al., 2020) as the SLM backbone for both SFT and FTC. In FTC, we use Fusion-in-Decoder (Izacard and Grave, 2020) to incorporate context information for decision-making. Specifically, each context related to a candidate answer is concatenated with the question and all candidate answers, then processed independently by the encoder. The concatenated representations of all contexts are subsequently fed into the decoder to generate predictions. In both datasets, we utilize AdamW (Loshchilov and Hutter, 2019) with a learning rate of $5 \times 10^{-5}$ for both SFT and FTC. We limit training to 100 epochs with a 200-step warm-up and apply early stopping after 5 epochs without validation improvement. The batch size is set to 8 for both datasets.

### A.7 PROMPT DETAILS

In this section, we present examples of prompts for both medical and general domains. The context of each example consists of three parts: (1) An overall context, which provides high-level information derived from the extracted keywords and candidate answers (red); (2) A specific context, which focuses on the knowledge associated with a candidate answer (blue) and its relation to the overall context (green); and (3) A preliminary decision, which draws a conclusion based on the contexts provided earlier (orange).

**Prompts for medical datasets.** Three clinicians were involved in the prompt design and writing. We held 5 meetings with clinicians to discuss the prompt design and iterate 4 versions. The final version of context needs around 10 minutes to write per context. Our medical prompts on *MedQA* and *MedMCQA* are based on Singhal et al. (2022), *HEADQA* is based on Wikipedia and written and verified by clinicians. Here we provide the prompts that we used in our experiments.

Table 13: Prompts for MedQA

---

**Question Keywords**: male, marathon runner, office, complaint, right-sided rib pain, Physical examination, normal heart, lung findings, exhalation, dysfunction, ribs 4-5, right, muscles, muscle groups, dysfunction, direct method
**Candidate Answers**: (a) anterior scalene (b) latissimus dorsi (c) pectoralis minor (d) quadratus lumborum

**Context**: Normal heart and lung findings on a physical exam coupled with evidence of exhalation dysfunction in ribs 4-5 on the right suggest a musculoskeletal cause of exertional chest pain.
(a): The anterior scalene muscle attaches to the first rib. It is not associated with exhalation dysfunction in ribs 4-5.
(b): The latissimus dorsi muscle attaches to ribs 9 and 10. It is not associated with exhalation dysfunction in ribs 4-5.
(c): The pectoralis minor muscle is attached to ribs 3, 4, and 5. Dysfunction in the fourth and fifth ribs can be caused by issues with the pectoralis minor muscle due to its attachment to these ribs. It is associated with exhalation dysfunction in ribs 4-5.
(d): Quadratus lumborum muscle attaches to ribs 11 and 12. It is not associated with exhalation dysfunction in ribs 4-5.
Therefore, the answer is (c).

**Question Keywords**: male, office, low back pain, denies, any, trauma, says, truck, day, job, Examination, patient, prone position, deep sacral, left, posterior inferior lateral angle, right, lumbosacral junction, springs, freely, compression, diagnosis
**Candidate Answers**: (a) left-on-left sacral torsion (b) left-on-right sacral torsion (c) right unilateral sacral flexion (d) right-on-right sacral torsion

**Context**: The physical exam shows the deep sacral sulcus on the left, a posterior inferior lateral angle on the right and normal spring test.
(a): This condition is characterized by the deep sacral sulcus on the right, a posterior inferior lateral angle on the left and normal spring test. It is not consistent with the findings from the physical exam.
(b): The left-on-right sacral torsion would be indicated by a deep sacral sulcus on the right, a posterior inferior lateral angle on the left, and a positive spring test. It is not consistent with the findings from the physical exam.

---

Table 13 – *Continued from previous page*

(c): This condition is characterized by a posterior inferior lateral angle on the right, a deep sacral sulcus on the right, and an absence of normal spring test. It is not consistent with the findings from the physical exam.

(d): This condition is characterized by a deep sacral sulcus on the left, a posterior inferior lateral angle on the right and normal spring test. It is consistent with the findings from the physical exam. Therefore, the answer is (d).

**Question Keywords**: man, comes, office, nonproductive cough, runny nose, frontal headache, headache, morning, nd, ibuprofen, relief, not, shortness of breath, Medical history, no medications, ibuprofen, pain, Vital signs, temperature, 37.4, °, 99.4, °, pulse, 88/min, 18/min, blood pressure, 120/84, Examination, nares, erythematous, mucous membranes, Examination, throat, erythema, follicular lymphoid hyperplasia, posterior oropharynx, no, cervical adenopathy, Lungs, clear, auscultation, patient's, symptoms
**Candidate Answers**: (a) Allergic rhinitis (b) Epstein-Barr virus (c) Mycoplasma pneumonia (d) Rhinovirus

**Context**: Sore throats are common symptoms in multiple upper respiratory viruses.
(a): A non-productive cough is a common symptom in upper respiratory viruses but is not present in allergic rhinitis. It is not the cause of the symptoms.
(b): The absence of shortness of breath indicates mycoplasma is less probable. It is not the cause of the symptoms.
(c): Cervical adenopathy is commonly seen in cases of Epstein Barr virus. The absence of cervical adenopathy indicates Epstein Barr virus is less likely. It is not the cause of the symptoms.
(d): Rhinovirus can cause this patient's symptoms, including sore throat, runny nose and a frontal headache. It is the cause of the symptoms.
Therefore, the answer is (d).

**Question Keywords**: healthy, woman, comes, physician, 8, months, husband, killed, car crash, decreased, difficulty falling asleep, states, sad, cries, frequently, door lock, five, house, five, pieces, toilet paper, perfectionist, urges, rituals, Pharmacotherapy, neurotransmitters
**Candidate Answers**: (a) Dopamine (b) Glutamate (c) Norepinephrine (d) Serotonin

**Context**: The woman is exhibiting symptoms of major depressive episodes, such as difficulty falling asleep, frequent crying, and a persistent feeling of sadness.
(a): Dopamine is a neurotransmitter that increases positive emotions. It is implicated in many disease processes, including Parkinson's and ADHD, and is targeted by antipsychotic medications but not used as a sleep aid. It is not a treatment for the patient's symptoms.
(b): Glutamate is a neurotransmitter that is associated with multiple neurological disorders including epilepsy, stroke, and autism. It is not a treatment for the patient's symptoms.
(c): Norepinephrine is a catecholamine with adrenergic properties. It is not a treatment for the patient's symptoms.
(d): Serotonin is a neurotransmitter which is the target for multiple antidepressants, anxiolytics, and antipsychotics. It could be a treatment to address the patient's symptoms of depression and anxiety. Therefore, the answer is (d).

**Question Keywords**: man, comes, office, preoperative, evaluation, adrenalectomy, scheduled, 2, weeks, One, month, care, emergency department, pain, right flank, motor vehicle collision, blood pressure, 160/100, mm Hg and CT scan, abdomen, incidental, left adrenal mass, laboratory studies, complete blood count, serum electrolyte concentrations, liver function tests, reference ranges, patient, healthy, elevated blood pressure, no medications, follow-up visit, office 2, weeks, disclosed, elevated, urinary normetanephrine, metanephrine, plasma, concentrations, patient, surgeon, recommended, adrenalectomy, vital signs, temperature, 36.6, 97.9, pulse, 100/min, 14/min, blood pressure, 170/95, Physical examination, no significant, findings, preoperative, preparation, treatment
**Candidate Answers**: (a) Labetalol (b) A loading dose of potassium chloride (c) Nifedipine (d) Phenoxybenzamine

Table 13 – *Continued from previous page*

**Context**: The patient is being evaluated for adrenalectomy due to a large left adrenal mass, which is likely causing elevated blood pressure as a symptom of pheochromocytoma. Elevated urinary normetanephrines confirm the diagnosis.
(a): This beta-blocker works by blocking the effects of adrenaline and other stress hormones on the heart and blood vessels. It is not a treatment for pheochromocytoma.
(b): The use of a potassium chloride loading dose is a treatment specifically for hypokalemia, which is a condition where there are abnormally low levels of potassium in the blood. It is not a treatment for pheochromocytoma.
(c): This drug is commonly prescribed to treat high blood pressure and angina. It can also help relieve symptoms of Raynaud's phenomenon. It is not a treatment for pheochromocytoma.
(d): This medication is used as a preoperative preparation treatment to block alpha-adrenergic receptors in the body and it effectively treats hypertension caused by pheochromocytoma. It is a treatment for pheochromocytoma.
Therefore, the answer is (d).

Table 14: Prompts for HEADQA

**Question Keywords**: autosomal dominant trait
**Candidate Answers**: (a) The trait appears more frequently in males. (b) The unaffected people do not transmit the trait. (c) The trait tends to skip generations. (d) The affected people have both affected parents. (e) The trait tends to appear in the progeny of related parents.

**Context**: Autosomal dominant inheritance is a mode of genetic transmission in which a trait or condition can be passed down from parent to child. One copy of a mutated gene from one parent can cause the genetic condition. For example, let 'A' represent the affected allele and 'a' represent the unaffected allele. An affected person may have the genotype AA or Aa, while an unaffected person has the genotype aa. Consequently, an individual with genotype AA has a 100% chance of passing on the affected allele, and someone with genotype Aa has a 50% chance of doing so.
(a): Autosomal dominant inheritance is not influenced by an individual's sex, as it is not sex-dependent. The expression of the trait occurs regardless of gender. It is not a characteristic of autosomal dominant inheritance.
(b): Unaffected individuals do not have the mutated gene and therefore cannot transmit the trait. It is a characteristic of autosomal dominant inheritance.
(c): Autosomal dominant traits can be passed down through multiple generations. Since the affected allele is dominant, an individual will express the trait as long as they inherit the affected gene. It is not a characteristic of autosomal dominant inheritance.
(d): Only one affected parent is needed to transmit on the autosomal dominant trait to their child. It is not a characteristic of autosomal dominant inheritance.
(e): A dominant gene can appear in any progeny, regardless of the parent. It is not a characteristic of autosomal dominant inheritance.
Therefore, the answer is (b).

**Question Keywords**: caring, patient, supraglottic laryngectomy
**Candidate Answers**: (a) He has lost the ability to speak by extirpation of the true vocal cords. (b) The tracheostomy they have performed will be permanent. (c) You have a risk of bronchoaspiration due to difficulty swallowing. (d) You may have constipation due to cervical dissection. (e) A portion of the larynx has been removed along with a vocal cord.

**Context**: Supraglottic laryngectomy or horizontal partial laryngectomy is an operation to remove the epiglottis, false vocal cords, and superior half of the thyroid cartilage.
(a): Supraglottic laryngectomy removes the false vocal cords, but the true vocal cords are not affected, and the patient's ability to speak should not be significantly impacted. It is not typical to lose the ability to speak by extirpation of the true vocal cords.
(b): If a tracheostomy tube is in place after the procedure, it is typically removed within 24-48 hours of surgery. It is not typical to involve A permanent tracheostomy as a part of the supraglottic laryngectomy process.

*Continued on next page*

(c): Supraglottic laryngectomy results in severe disturbance to the swallowing mechanism by removal of protective layers and sensation. There is an increased risk of bronchoaspiration. It is related to care of patients with supraglottic laryngectomy.

(d): Constipation is not a side effect of supraglottic laryngectomy. It is not related to care of patients with supraglottic laryngectomy.

(e): Supraglottic laryngectomy is an operation to remove the epiglottis, false vocal cords, and superior half of the thyroid cartilage. In this procedure, the true vocal cords are not typically affected, preserving the patient's ability to speak as much as possible. It is not common to remove a portion of the larynx along with a true vocal cord during this procedure.

Therefore, the answer is (c).

**Question Keywords**: estrogenic treatment, adverse effects, NOT, adverse effect, pharmacological action

**Candidate Answers**: (a) Edema (b) Breast pain (c) Ovarian cancer (d) Sickness (e) Headaches

**Context**: Estrogen therapy involves supplementing a patient with estrogen, the primary female sex hormone. Potential side effects include breast tenderness or swelling, edema, nausea, leg cramps, endometrial cancer, and more.

(a): Edema is a potential adverse effect of estrogen therapy. Estrogen and aldosterone both originate from cholesterol, and an excessive amount of estrogen in the body can stimulate aldosterone receptors, leading to water retention in nephrons. This water retention can result in edema. It is a non-adverse effect.

(b): Estrogen promotes ductal growth and fat deposition in the breasts. Excessive estrogen levels can lead to mammary duct hyperplasia, which may result in breast pain. It is not a non-adverse effect.

(c): Ovarian cancer is not known to be an adverse effect of estrogenic treatment. It is a possible choice for a non-adverse effect.

(d): Edema is a possible adverse effect of estrogenic treatment, and swelling in body parts may cause the feeling of sickness. It is not a non-adverse effect.

(e): Headache is a possible adverse effect of estrogenic treatment. It is not a non-adverse effect.

Therefore, the answer is (c).

**Question Keywords**: cardiac valvular prosthesis, biological, mechanical, implanted, patient, aspects, characteristics, patient, prosthesis, INCORRECT, statement

**Candidate Answers**: (a) Permanent anticoagulation is necessary in mechanical prostheses. (b) In general, biological prostheses are indicated in young patients, with long life expectancy. (c) Biological prostheses would be indicated in cases that present a formal contraindication for anticoagulation. (d) The rate of structural deterioration of a biological prosthesis is inversely proportional to the age of the subject. (e) Biological prostheses do not require permanent anticoagulation.

**Context**: Cardiac valvular prostheses (biological or mechanical) are artificial cardiac valves implanted into a patient's heart. Mechanical valves may last a lifetime, but they come with an increased risk of blood clots, necessitating the use of blood thinners such as warfarin. In contrast, biological valves, which are made from pig or cow tissue, do not increase the risk of bleeding or clotting but tend to wear out sooner.

(a): Mechanical valves increase the risk of blood clotting. It is not an incorrect statement.

(b): The latest revisions of the ESC/EACTS guidelines suggest that bioprostheses are acceptable in patients aged between 60 and 65 years at the time of surgery. The reoperation rate for structural valve degeneration (SVD) of bioprostheses occurred exclusively among patients younger than 56 years. Young patients are not typically recommended for a biological prosthesis.

(c): Biological prostheses, which are made from pig or cow tissue, do not increase the risk of either bleeding or clotting but will wear out sooner. It is not an incorrect statement.

(d): The disadvantages of biological heart valves are a smaller valve orifice area and the risk of structural valve degeneration, which may necessitate reoperation. Thus, the younger the patient, the higher risk of structural deterioration. It is not an incorrect statement.

(e): Biological prosthesis do not increase the risk of clotting so do not require permanent anticoagulant. It is not an incorrect statement.

Therefore, the answer is (b).

Table 14 – *Continued from previous page*

**Question Keywords**: connection, automatic, emotional responses, control, behaviors, guiding, behavior, manifestation, emotional responses

**Candidate Answers**: (a) The angular gyrus of the limbic system. (b) The convolution or lobe of the insula. (c) The prefrontal orbitofrontal or ventromedial cortex. (d) The thalamus (e) The cortex of somatosensory association.

**Context**: The prefrontal orbitofrontal cortex has multiple functions including mediating context specific responding, encoding contingencies in a flexible manner, encoding value, encoding inferred value, inhibiting responses, learning changes in contingency, emotional appraisal, altering behavior through somatic markers, driving social behavior, and representing state spaces. The orbitofrontal cortex thus plays a key role in emotion, by representing the reward value of the goals for action.
(a): The angular gyrus (AG) is a hub of several networks that are involved in various functions, including attention, self-processing, semantic information processing, emotion regulation, and mentalizing. It is not the area responsible for connecting automatic emotional responses and controlling complex behaviors.
(b): The insula is important for gustatory and sensorimotor processing, risk-reward behavior, autonomics, pain pathways, and auditory and vestibular functioning. It is not the area responsible for connecting automatic emotional responses and controlling complex behaviors.
(c): The prefrontal cortex guides behavior by controlling the manifestation of emotional responses through understanding rewards, encoding values, and driving behaviors. It is the potential correct answer.
(d): The thalamus acts as the body's information relay station. All sensory information (except for olfaction) must be processed through the thalamus before being sent to the cerebral cortex for interpretation. It is not the area responsible for connecting automatic emotional responses and controlling complex behaviors.
(e): The somatosensory cortex is responsible for processing all bodily sensations. These sensations originate from receptors located throughout the body that detect temperature, pain, touch, pressure, and proprioception. It is not the area responsible for connecting automatic emotional responses and controlling complex behaviors.
Therefore, the answer is (c).

Table 15: Prompts for MedMCQA

**Question Keywords**: Maximum, increase, prolactin level

**Candidate Answers**: (a) Risperidone (b) Clozapine (c) Olanzapine (d) Aripiprazole

**Context**: The four drugs in answer choices are all atypical antipsychotics, which are used to treat psychotic conditions like schizophrenia through blockage of dopamine and serotonin receptors. These drugs block dopamine D2 receptors and serotonin 5-HT2 receptors. Maximum increase in prolactin, or hyperprolactinemia, is one of the side effects of atypical antipsychotics, because dopamine tends to inhibit prolactin release from the anterior pituitary. (a): Risperidone is a type of atypical antipsychotics that block dopamine D2 receptor and serotonin 5-HT2 receptor. It is generally used to treat schizophrenia or disorders with concomitant psychosis. Hyperprolactinemia is one of the most common side effects of risperidone. It is the drug to increase prolactin levels.
(b): Clozapine is used to treat schizophrenia or disorders with concomitant psychosis. Clozapine is associated with side effects such as agranulocytosis, seizures, and myocarditis, but it does not appear to elevate prolactin levels. It is not the drug to increase prolactin levels.
(c): Olanzapine is used to treat schizophrenia or disorders with concomitant psychosis. The side effect of olanzapine does not include hyperprolactinemia. It is not the drug to increase prolactin levels.
(d): Aripiprazole is generally used to treat schizophrenia or disorders with concomitant psychosis. The side effect of olanzapine does not include hyperprolactinemia. It is not the drug to increase prolactin levels.
Therefore, the answer is (a).

Table 15 – *Continued from previous page*

**Question Keywords**: male, complains, severe back pain, inability, left lower limb, Radiographic studies, compression, nerve elements, intervertebral, foramen, vertebrae L5, S1, structure, space-occupying lesion

**Candidate Answers**: (a) Anulus fibrosus (b) Nucleus pulposus (c) Posterior longitudinal ligament (d) Anterior longitudinal ligament

**Context**: The male is complained of a severe back pain and inability to move, and radiographic evidence shows the compression of a nerve component. This may suggest a herniated intervertebral disk through a tear in the surrounding annulus fibrosus. The soft, gelatinous nucleus pulposus is forced out through a weakened part of the disk, compressing nerve components of the spinal cord and resulting in back pain and nerve root irritation. This impingement is resulting in paralysis, and should be considered a medical emergency.
(a): Annulus fibrosus is a tough, circular exterior of the intervertebral disc, made up of fibrous connective tissue. It surrounds the soft inner core, the nucleus pulposus. It is not the component that is forced out by the tear.
(b): Nucleus pulposus is the inner core of the vertebral disc. The tear in the annulus fibrosus causes it to be forced out. It could result in compression of the nerve components of the vertebrae.
(c): Posterior longitudinal ligament connects and stabilizes the bones of the spinal column. It runs almost the entire length of the spine, from the 2nd vertebra in the cervical spine (neck) all the way down to the sacrum (end of the spine). This ligament is located adjacent to the spinal cord. It is not easily teared or curved.
(d): Anterior longitudinal ligament is a ligament that runs down the anterior surface of the spine. It traverses all of the vertebral bodies and intervertebral discs on their ventral side. It has a high tensile strength and is resistant to tearing or deformation. It is not easily teared or curved.
Therefore, the answer is (b).

**Question Keywords**: Neuroendocrine cells, lungs

**Candidate Answers**: (a) Dendritic cells (b) Type I pneumocytes (c) Type II pneumocytes (d) APUD cells

**Context**: Neuroendocrine cells are part of the neuroendocrine system. The neuroendocrine cells of the lung make hormones that control the flow of air and blood in the lungs.This may suggest a herniated intervertebral disk through a tear in the surrounding annulus fibrosus. The soft, gelatinous nucleus pulposus is forced out through a weakened part of the disk, compressing nerve components of the spinal cord and resulting in back pain and nerve root irritation. This impingement is resulting in paralysis, and should be considered a medical emergency.
(a): Dendritic cells are a type of antigen-presenting cell in the immune system that act as messengers between the innate and adaptive immune systems. It is not a type of neuroendocrine cell.
(b): Type I pneumocytes are alveolar cells that line the alveolar surface of the lungs and are responsible for gas exchange. It is not a type of neuroendocrine cell.
(c): Type II pneumocytes are alveolar cells that secrete surfactant to reduce alveolar surface tension and prevent alveolar collapse. It is not a type of neuroendocrine cell.
(d): APUD cells are a type of neuroendocrine cell that function through amine precursor uptake and decarboxylation. It is accurate to say that they are a type of neuroendocrine cell.
Therefore, the answer is (d).

**Question Keywords**: Presence, remote, contamination,water

**Candidate Answers**: (d) Streptococci (b) Staphalococci (c) Clastridium pertringes (d) Vibrio

**Context**: Infections that can be spread through water contamination are generally transmitted orally or via fecal matter. (a): Streptococci are spread through direct contact with the nose and throat discharges of an infected individual or with infected skin lesions. Water is not a medium for the spread of streptococci. It is not related water contamination.
(b): Staphylococci is spread by skin contact, like a bite or cut. It is not related water contamination.
(c): Clostridium perfringens are one of the most common causes of food poisoning. They are environmentally stable and specific to contamination by sewage. Their spread is a indicator of water contamination.
(d): Vibrio species are gram-negative bacteria that spread through foodborne infection, but they are highly salt tolerant and unable to survive in fresh water. It is not related water contamination.

*Continued on next page*

Table 15 – *Continued from previous page*

Therefore, the answer is (c).

**Question Keywords**: True, Mooren's ulcer, 2007, 2013
**Candidate Answers**: (a) Painless condition (b) Affects cornea (c) Sudden loss of vision (d) Bilateral in majority of cases

**Context**: Mooren's ulcer is characterized by painful peripheral corneal ulceration of unknown etiology. The disease generally begins with intense limbal inflammation and swelling in the episclera and conjunctiva. Patients often experience severe pain, photophobia, and tearing along with a red inflamed eye.
(a): Mooren's ulcer is a painful ulceration of the eye. It is not the truth of Mooren's ulcer.
(b): Mooren's ulcer is characterized by painful peripheral corneal ulceration of unknown etiology. It is the truth of Mooren's ulcer.
(c): The symptoms of Mooren's ulcer do not include sudden loss of vision. It is not the truth of Mooren's ulcer.
(d): About one third of Mooren's ulcer cases present bilaterally. The proportion is less than half. It is not the majority of cases.
Therefore, the answer is (b).

**Prompts for general domain datasets.** Our prompts on *CommonsenseQA* and *OpenbookQA* are based on (Li et al., 2022).

Table 16: Prompts for Commonsense QA

**Question Keywords**: fountain pen, people, ink, absorb, pen, hand done, extra, use, fountain
**Candidate Answers**: (a)shirt pocket (b) calligrapher's hand (c) inkwell (d) desk drawer (e) blotter

**Context**: Fountain pens need to be filled with ink for writing. Extra ink should be absorbed using special tools.
(a): A fountain pen can be conveniently carried in a shirt pocket. It is not associated with the tool to absorb extra ink from fountain pens.
(b): Calligraphers use fountain pens to create stunning handwriting. It is not associated with the tool to absorb extra ink from fountain pens.
(c): An inkwell serves as a container for the ink used in a fountain pen. It is not associated with the tool to absorb extra ink from fountain pens.
(d): A fountain pen can be kept safely in a desk drawer. It is not associated with the tool to absorb extra ink from fountain pens.
(e): Blotters are designed to absorb excess ink from pens. It is the tool for absorbing extra ink.
Therefore, the answer is (e).

**Question Keywords**: fox, forest, walk, look, city
**Candidate Answers**: (a) pretty flowers (b) hen house (c) natural habitat (d) storybook (e) dense forest

**Context**: Foxes are animals that typically live in forests. They walk from the city to the forest to look for their living place.
(a): Pretty flowers are in forests. It is not a reason for a fox walking into the forest.
(b): Foxes sometimes prey on chickens in hen houses. It is not a reason for a fox to walk into the forest.
(c): Forests are the natural habitat of foxes. Foxes walk from city to forest to look for their natural habitat. (d): Forests and foxes are common subjects in storybooks. It is not a reason for fox walking to the forest.
(e): Dense forest is a type or category of forests characterized by having a high density of trees and vegetation. It is a type of forest.
Therefore, the answer is (c) or (e).

**Question Keywords**: grape, put, check

*Continued on next page*

Table 16 – *Continued from previous page*

**Candidate Answers**: (a) mouth (b) grocery cart (c) super market (d) fruit basket (e) fruit market

**Context**: Grapes need to be put into a place for checking out.
(a): Grapes can be eaten by mouth. It is not a place to put grapes for checking out.
(b): Grapes can be brought during grocery shopping and people put groceries into grocery carts before checking out. It could be a potential place to put grape.
(c): Super markets sell grapes. It is not a place to put grapes for checking out.
(d): Fruit markets sell grapes. It is not a place to put grapes for checking out.
(e): Fruit baskets are often used as gifts to hold and present a variety of fresh grapes. It is not a place to put grapes for checking out.
Therefore, the answer is (b).

**Question Keywords**: drawstring bag, head, woman, bag, drawstring, check, baggage
**Candidate Answers**: (a) garbage can (b) military (c) jewelry store (d) safe (e) airport

**Context**: A woman can check baggage such as a drawstring bag at the check-in counter.
(a): A garbage can is a container that is specifically designed to hold and contain trash or waste materials. It is not related to the context.
(b): Military refers to the armed forces of a country, which is responsible for defending the nation and its interests against external threats. It is not a place where a woman can check bags.
(c): Jewelry stores sell jewelry. It is not a typical place to check baggage.
(d): Check baggage could keep the bag safe. A woman can check her drawstring bag to keep the bag safe.
(e): Airport is a place where the woman can check her drawstring bag as baggage at the check-in-counter. It is common to check baggage in airport.
Therefore, the answer is (e).

**Question Keywords**: cable, entertainment, home, require, equipment
**Candidate Answers**: (a) radio shack (b) substation (c) television (d) cabinet (e) desk
**Context**: A cable transmits electricity or information and data to home entertainment equipment that requires electricity.
(a): Radio Shack is a retailer that sells cable. It is not a home entertainment equipment used cable.
(b): Cables are used to transmit electrical energy between substations and other parts of the electrical power system. It is not a home entertainment equipment used cable.
(c): Television is a type of home electric entertainment equipment that requires cable. It is a home entertainment equipment used cable.
(d): Cabinet is a place to store cable. It is not a home entertainment equipment used cable.
(e): Desk with built-in cable management features can help keep cables tidy. It is not a home entertainment equipment used cable.
Therefore, the answer is (c).

**Question Keywords**: people, populate, might, may, sammy, go
**Candidate Answers**: (a) populated areas (b) race track (c) desert (d) apartment (e) roadblock

**Context**: People may like to go to places where people populate together. (a): Populated areas are locations where people gather and live in close proximity to each other. It could be a place where people populate together. (b): Deserts are inhospitable environments for people. It is not a place where people populate together. (c): People go to race competitions on the race track. It could be a place where people populate together. (d): Apartments serve as living spaces for people. It is not a place where people populate together. (e): Roadblocks are structures set up to restrict or regulate the movement of people and vehicles. It is not a place where people populate together. Therefore, the answer is (a) or (c).

**Question Keywords**: highway, maps, replace, street, google, map, highway, gps, service
**Candidate Answers**: (a) united states (b) mexico (c) countryside (d) atlas (e) oceans

**Context**: Google Maps and GPS services have replaced traditional physical maps for navigating highways and streets.
(a): People in the United States use Google Maps and GPS services to navigate highways and streets. It is not the tool that GPS replaced with.

*Continued on next page*

Table 16 – *Continued from previous page*

(b): People in Mexico use Google Maps and GPS services to navigate highways and streets. It is not the tool that GPS replaced with.

(c): Google Maps and GPS services cover the countryside. It is not the tool that GPS replaced with.

(d): Google Maps and GPS services have replaced traditional physical maps for navigating highways and streets. Atlases are examples of traditional physical maps. It is the tool that GPS replaced with.

(e): Google Maps and GPS services cover the oceans and are commonly used in marine navigation. It is not the tool that GPS replaced with.

Therefore, the answer is (d).

Table 17: Prompts for Openbook QA

**Question Keywords**: acid, environment, aquatic, rain, effect, acid rain

**Candidate Answers**: (a) decrease in plant life (b) increase in fish population (c) increase in plant growth (d) cleaner and clearer water

**Context**: The acid rain is a type of rain that has an acidic effect due to the presence of acid in the atmosphere. Acid rain is harmful to the environment, especially aquatic life. The acid in the rain can have a negative effect on the water quality of aquatic environments.

(a): Acid rain can have a negative effect on plant life. The acid in the rain can damage plant cells and cause a decrease in plant growth, leading to a decrease in plant life. It is likely to have a decrease in plant life by acid rain.

(b): Acid rain can have a harmful effect on aquatic life, including fish. The acid in the water can make it difficult for fish to breathe and can harm their reproductive systems. It is not likely to have an increase in fish population by acid rain.

(c): As previously mentioned, the acid in the rain can damage plant cells and cause a decrease in plant growth. It is not possible to have an increase in plant growth.

(d): Acid rain can have a harmful effect on water quality, making it more acidic and harmful to aquatic life. It is not possible to have cleaner and clearer water.

Therefore, the answer is (a).

**Question Keywords**: moon, surface

**Candidate Answers**: (a) is smooth on the entire surface (b) contains large cavities cause by explosions (c) contains an internal core of cheese (d) is filled with lakes

**Context**: The moon is a natural satellite that orbits around the Earth. Its surface is covered with dead volcanoes, impact craters, and lava flows, some visible to the unaided stargazer.

(a): The moon has mountains, craters, and other features caused by impacts from meteoroids and asteroids. It is not entirely smooth on the surface.

(b): Impact craters are formed when an asteroid craters, each of which was formed when an asteroid or comet collided with the Moon's surface. The moon's surface contains large cavities caused by explosions from impacts.

(c): The core is largely composed of iron and some nickel. The inner core is a solid mass about 480 km in diameter. It does not contain an internal core of cheese. (d): The moon has lunar maria composed of basalt formed from surface lava flows that later congealed. It is not filled with lakes.

Therefore, the answer is (b).

**Question Keywords**: car, approach, night

**Candidate Answers**: (a) the headlights become more intense (b) the headlights recede into the dark (c) the headlights remain at a constant (d) the headlights turn off

**Context**: Headlights of a car are a source of light. As a car approaches, the source of light becomes closer, and that source will appear brighter.

(a): As the car becomes closer, the distance to the source of light decreases. The headlights become brighter and more intense. This is a possible phenomenon.

(b): If the source does not change and the headlights are closer, the headlights cannot become dimmer. This is not a commonsense relation.

*Continued on next page*

*Table 17 – Continued from previous page*

(c): If the distance to the source of light changes, the brightness of headlights will change. It is not able to remain constant.
(d): Turning off the headlights would cause the driver to be driving in complete darkness, which is dangerous and can lead to accidents. It is not a reasonable condition.
Therefore, the answer is (a).

**Question Keywords**: change, easter, weather change, weather, christmas
**Candidate Answers**: (a) the air may chill (b) the ground may freeze (c) the plants may die (d) the ground may warm
**Context**: In the US, Christmas falls in the winter season, while Easter arrives at the beginning of spring.
(a): The air becomes chill as temperature drops. The temperature commonly increases from winter to spring. It is not a likely scenario.
(b): During winter, the ground usually freezes, whereas in spring, it does not. It is not a probable scenario.
(c): Extreme cold or hot weather can cause plants to die. The beginning of spring provides suitable weather conditions for plants to grow. It is not common to have plants die.
(d): As winter transitions into spring, the weather becomes warmer. The temperature of the ground is influenced by the weather.
Therefore, the answer is (d).

**Question Keywords**: heat, recipe, moisture, good, ocean
**Candidate Answers**: (a) a violent storm (b) violent sea animals (c) condensation (d) inland storms
**Context**: The ocean, a vast body of water that covers a large portion of the Earth's surface, serves as a source of heat and moisture.
(a): The heat and moisture present in the ocean can create ideal conditions for a hurricane or typhoon. Hurricane and typhoon are violent storms.
(b): Violent sea animals are not related to heat and moisture in the ocean. It is not a likely choice.
(c): Condensation is the process by which water vapor becomes liquid, which is the reverse of evaporation. This can happen in one of two ways: either the air is cooled to its dew point or it becomes so saturated with water vapor that it cannot hold any more water. It is not likely to occur in hot conditions.
(d): Although heat and moisture can cause inland storms, they are not directly related to the ocean. It is not a likely choice.
Therefore, the answer is (a).

**Question Keywords**: hummingbird, take
**Candidate Answers**: (a) bees (b) energy (c) pollen (d) honey
**Context**: Hummingbirds dip their long bills into flowers to drink nectar to get energy.
(a): Hummingbirds and bees are both attracted to the sweet nectar produced by flowers, but bees extract the nectar from the base of the flowers, while hummingbirds dip their long bills into the flowers to drink the nectar and obtain energy. No relationship can be found.
(b): Hummingbirds obtain energy by getting nectar from flowers through dipping their long bills into the flowers. No relationship can be found.
(c): When hummingbirds drink nectar, they also inadvertently take grains of pollen which stick to their feathers and bills, and get carried to the next flower they visit. No relationship can be found.
(d): Hummingbirds do not produce or consume honey. This fact is unrelated to their method of obtaining energy by drinking nectar from flowers.
Therefore, the answer is None.

**Question Keywords**: responsible, sun
**Candidate Answers**: (a) puppies learning new tricks (b) children growing up and getting old (c) flowers wilting in a vase (d) plants sprouting, blooming and wilting
**Context**: The sun is the source of energy for physical cycles on Earth.

*Continued on next page*

Table 17 – *Continued from previous page*

(a): Puppies learning new tricks involves the acquisition and processing of information, which is essential for the puppies to learn and adapt to their environment. It is not directly related to the effect of the sun.

(b): Children grow up and age over time. The sun is not directly responsible for the passage of time itself. It is not directly related to the effect of the sun.

(c): Flowers in a vase become wilting because they are cut from their original source of nutrients and water and are no longer able to receive the essential nourishment they need to stay healthy and vibrant. It is not directly related to the effect of the sun.

(d): Plants need sunlight to photosynthesize and grow, and the sun's heat and light play a crucial role in the process of plant growth and decay. It is the thing that the sun is responsible for.

Therefore, the answer is (d).

