# OpenReview forum: "Enhancing Small Medical Learners with Privacy-preserving Contextual Prompting"
_ICLR.cc/2024/Conference — ICLR 2024 poster_

### Official Review · Reviewer_JSi7 · 2023-10-30

**Soundness:** 3 good
**Presentation:** 3 good
**Contribution:** 3 good
**Rating:** 6
**Confidence:** 3

**Summary:**

This article discusses a method to improve the application of SLM in the medical field, utilizing LLM's medical proficiency to boost SLM performance in medical tasks under privacy-restricted scenarios which has important social significance. The method was tested on MedQA, HEADQA, MedMCQA, and MMLU-professional medicine datasets, showing some improvements over existing methods. Additionally, the authors compared results across different sizes of training sets.

**Strengths:**

see summary

**Weaknesses:**

1). Imprecise example of Privacy Protection.
The example in Figure 1 indicates that personal privacy issues are only present in the first sentence, and the key words "man" and "admitted" in that sentence have almost no impact on the subsequent content. Could it then be possible to simply delete the first sentence to achieve privacy protection, as extracting key words here does not seem to play a significant role.

2). Privacy Protection as an Innovation Point
Regarding the extraction of key words for privacy protection, the paper uses a medical NER model proposed by Neumann et al in 2019. We suggest further improvement of this model, for example, considering age as a crucial keyword for certain diseases and extracting it as necessary to better enrich the innovative aspects of the paper.

3). Ambiguity of Symbols in Annotations
Annotation 13 on page 8 only appears in the content of the article but is not explained.

4) The overall innovation of the methodology needs improvement, as the majority of the content relies on existing methods, such as the medical NER (Named Entity Recognition) model.

**Questions:**

please see the weaknesses.

---

> ### Author Response · Authors · 2023-11-18
>
> Thank you for your thorough review and valuable feedback. Please find our responses to your concerns below.
>
> **1. Concerns about the novelty.**
>
> We appreciate the opportunity to further elucidate the innovative aspects of our work. Our paper introduces a novel approach that simplifies the integration of medical knowledge into SLMs by leveraging LLMs as a knowledge base. Unlike prior research that often relies on complex training algorithms and the intricate process of constructing medical knowledge graphs[1-3], our method streamlines this integration, which is substantiated by the performance improvements showcased in Table 1 in the paper.
>
> Moreover, our paper begins a crucial discussion on privacy issues when using LLMs in healthcare. This is a significant issue that hasn't been given enough focus in past studies [4-7]. By bringing this up, we aim to encourage more careful and ethical use of LLMs where patient privacy is a concern.
>
> **2. Concerns on using NER.**
>
> In our paper, we utilize NER to extract keywords automatically, and we mention in the introduction that keywords can also be extracted by other methods, e.g., a manually created dictionary based on domain expertise. In practice, we can also use rules to post-process the extracted keywords or  keep clinicians in the loop to add or delete medical terms, thereby improving the quality of the medical keywords.
>
> We adopt NER here because it is the most convenient way to extract relatively high-quality medical information from raw data. This supports our main idea that by utilizing LLMs as a database, we can generate medical knowledge for auxiliary SLMs in medical decision-making while reducing privacy concerns. We will further clarify the reason to utilize NER in our revision to improve clarity.
>
> **3. Suggestion about missed  footnote.**
>
> We are grateful for your attention to detail. We will add this footnote in the revision.
>
>
> [1] Yasunaga et al. (2022a) Qa-gnn: Reasoning with language models and knowledge graphs for question answering \
> [2] Zhang et al. (2022) Greaselm: Graph reasoning enhanced language models for question answering \
> [3] Yasunaga et al. (2022b) Deep bidirectional language-knowledge graph pretraining \
> [4] Ho et al., (2022) Large language models are reasoning teachers \
> [5] Shridhar et al., (2022) Distilling multi-step reasoning capabilities of large language models into smaller models via semantic decompositions \
> [6] Wang et al., (2022) Pinto: Faithful language reasoning using prompt-generated rationales \
> [7] Li et al., (2022) Explanations from large language models make small reasoners better

---

### Official Review · Reviewer_gXvF · 2023-11-01

**Soundness:** 3 good
**Presentation:** 2 fair
**Contribution:** 3 good
**Rating:** 6
**Confidence:** 4

**Summary:**

This paper tried to improve the performance of small medical language models by introducing knowledge from large language models, which keeps the privacy of clinical text when using large language models.  The proposed method uses keywords instead of full raw text to generate initial evidence from LLM and feed the evidence to small language model.

**Strengths:**

Privacy-preserving is an essential and common need when using LLM in clinical text. This paper tried to solve this problem by using keywords instead of raw text, the idea is novel and experiments demonstrated the effectiveness of this approach.

**Weaknesses:**

1. As this research utilized a named entity recognition model to extract keywords, it is possible that the NER model can extract privacy information such as patient names. Is there any filtering or postprocessing step to avoid that? In addition, it is not guaranteed that NER system will never extract sensitive patient information; for example, if the NER system incorrectly extracts a patient's address as a symptom, then the address may be leaked to LLM. Although it is very rare, it is still necessary to comment on this.
2. As the LLM already provides a preliminary decision, I am curious about the performance if we only feed the preliminary decision from LLM to SLM. It is worth knowing which part of the LLM-generated information improves the SLM most.
3. The related work section need to discuss more LLM application in the clinical area, especially the knowledge-enhanced LLM in clinical settings. For example, paper "Qualifying Chinese Medical Licensing Examination with Knowledge Enhanced Generative Pre-training Model." also utilized external knowledge for clinical questions.
4. By adding the LLM-generated content, will the new concatenated input be too long and out of the word window in SLM? How do you deal with the long content problem?

**Questions:**

By adding the LLM-generated content, will the new concatenated input be too long and out of the word window in SLM? How do you deal with the long content problem?

---

> ### Author Response · Authors · 2023-11-18
>
> We appreciate the insightful feedback and comments from the reviewer. Their positive observations about the novelty and thoroughness of our experiments are very encouraging. We have addressed your concerns in our response.
>
> **1. Concerns on privacy preserving in practical usage.**
> The data we utilized in experiments have already undergone post-processing; however, even well-processed data cannot be directly shared with third parties in a real-hospital setting. Here, we adopt NER methods directly, solely for automation, to show that LLM can be utilized as a medical database to query knowledge under privacy-restricted scenarios. Practically, we can leverage de-identification models and rules to remove personal information and then extract medical keywords to query third-party LLMs for auxiliary knowledge generation. In this paper, we take an initial step to discuss the significant privacy-preserving situations in the medical domain and demonstrate the promising results of utilizing LLM to improve SLM while mitigating privacy concerns.
>
> **2.Question about what SLM learns for decision making.**
>
> We feed preliminary decisions (PD) as context into SLM with backbone BioLinkBert-Base on three datasets. Three separate runs for each setting are conducted and the average results along with the standard deviation are reported. The results are shown in the Table below.
>
>
> |            | MEDQA         | HeadQA        | MEDMCQA       |
> |------------|---------------|---------------|---------------|
> | SLM w PD   | 47.21 ± 0.31  | 53.64 ± 1.09  | 45.42 ± 0.17  |
> | FTC        | 50.17 ± 0.42  | 61.35 ± 0.16  | 49.20 ± 0.45  |
>
> FTC, which integrates extensive medical knowledge into the decision-making, shows a consistent and significant improvement over the SLM that only uses PD for context. These findings underscore the valuable contribution of leveraging comprehensive medical knowledge, provided by LLM, in enhancing the medical decision-making capabilities.
>
> **3. Suggestion about related work in LLM application in the clinical domain.**
>
> Thanks for the suggestion in the related work. We will add the suggested work into the related work section in the revision.
>
> **4. Question abut address long medical context generated by LLM.**
>
> We utilize the Fusion-in-Decoder [1] approach in our general domain experiments. This strategy is also effective for encoding long contexts. It works by dividing the input into smaller passages, encoding each one separately, and then combining the encoded representations for decision-making.
>
> [1] Izacard et al. ( 2020) Leveraging passage retrieval with generative models for open domain question answering

---

> ### Comment · Reviewer_gXvF · 2023-11-22
> **Thanks**
>
> Thanks for replying. Based on the response, I would like to keep my original score.

---

### Official Review · Reviewer_TtE2 · 2023-11-01

**Soundness:** 2 fair
**Presentation:** 2 fair
**Contribution:** 3 good
**Rating:** 6
**Confidence:** 4

**Summary:**

The paper studied medical QA problems by incorporating large language models (LLMs) to assist small-language models (SLMs). To protect the private information in the data, the authors propose to first extract keywords and then use the keywords to query LLMs for intermediate content which can be used for SLMs to enhance prediction accuracy.

**Strengths:**

1. (originality) The proposed method is novel by extracting keywords and privately incorporating LLM for SLM-based predictions.
2. (clarity) Overall, the paper is fair in presentation. The demonstrations of synthetic medical data with private information and extracted keywords are helpful for understanding the concepts.
3. (significance) Versus the compared baselines, the proposed methods significantly improve the prediction accuracy on three medical QA tasks.
4. (quality) The authors thoroughly evaluate the performance of the proposed method.

**Weaknesses:**

1. (Clarity) There is no specific definition of the private information. From Figure 1, it seems that privacy definition is restricted to private identifiable information (PII). The authors should clarify the scope of privacy risks. Importantly, the proposed method cannot address general private information leakage that is considered by strict formulations like differential privacy.
2. (Quality) The evaluation of privacy is not strict.
  - Risks: It is possible that the keyword extraction includes private identifiable information (PII), for instance, names and dates as shown in Figure 1. There is no theoretical guarantee for privacy protection or empirical evaluation of the leakage rates of such PII.
  - Metric: The authors used the privacy budget for quantifying privacy risks:  the ratio of the number of words provided to the LLM to the total words in the original question. However, I doubt if the metric can imply some privacy risks. There essentially lacks an intuitive explanation of the relationship between the privacy budget and privacy risks.
3. (Motivation) As the authors said, SLM presents a large gap compared to LLMs and thus there is no clear motivation to use SLM for prediction. Although the authors mention that ChatGPT requires access to data, it is essentially ignored that open-source LLMs, for example, Llama, can be used. In the paper, there is no referred evidence for the large gap between open-source LLMs and ChatGPT on the concerned medical tasks. Thus, I strongly doubt if the motivation of the paper can hold.

**Questions:**

* There is no clear motivation to see SLM for prediction. Although the authors mention that ChatGPT requires access to data, it is essentially ignored that open-source LLMs, for example, Llama, can be used. Is there any evidence for the large gap between open-source LLMs and ChatGPT on the concerned medical tasks?

---

> ### Author Response · Authors · 2023-11-18
>
> We would like to thank the reviewer for their comments and feedback. We reply their concerns in the below.
>
> **1. Concern  on privacy definition.**
> Privacy is a multifaceted concept with varying definitions that depend on the context and use cases. Our definition of privacy differs from that of differential privacy, which offers a theoretical guarantee but may not always be practically applicable. In our setting, privacy issues do not arise during the training process but instead when calling third-party APIs with tokens to query medical knowledge. Specifically, we define the privacy budget as the ratio of the number of words from the original question that are provided to the third party for prompting the language model, as detailed in Section 5. Intuitively, the less information provided to the third party, the better the privacy condition we achieve; if no words are sent, there is no privacy risk.
>
> Furthermore, Figure 1 is for demonstration purposes only and is not used to define privacy in terms of personally identifiable information (PII). In practice, we can employ additional de-identification methods to remove PII from medical terms in the preprocessing or postprocessing stages. In this paper, our primary contribution is to demonstrate the effectiveness of using LLM as a database for generating medical knowledge to enhance SLMs' performance. We also initiate the discussion of LLM utilization under privacy-restricted conditions, which previous works [1-4] have overlooked. We will revise our paper to include a clear definition of privacy in our context, thereby improving its clarity.
>
>
> **2.  Concerns about motivation to utilize ChatGPT instead of open-source LLMs.**
>
> We evaluate medical domain specific LLMs based on LLaMA [5] : AlpaCare [6], PMC_LLAMA [7], ChatDoctor [8], Medalpaca [9] and Baize-healthcare [10]  on the three QA datasets following [11]. The results are shown in the table below.
>
> |               | MEDQA  | HeadQA | MEDMCQA |
> | ------------- | ------ | ------ | ------- |
> | FTC           | 53.85  | 63.17  | 52.09   |
> | ChatGPT-privacy | 41.7  | 47.5   | 35.2    |
> | AlpaCare      | 35.0   | 29.2   | 33.5    |
> | PMC_LLAMA     | 34.2   | 28.1   | 28.6    |
> | ChatDoctor    | 34.3   | 30.0   | 33.5    |
> | MedaIpaca     | 38.4   | 30.3   | 31.3    |
> | Baize-healthcare | 34.5 | 29.3   | 32.5    |
>
>
> ChatGPT significantly outperforms domain-specific LLMs across three QA datasets under privacy-restricted settings, demonstrating that the medical knowledge within open-source LLMs still lags behind the GPT-series models. Additionally, our FTC, which further leverages the knowledge provided by ChatGPT, achieves better performance, indicating the effectiveness of our methods.
>
>
> [1] Ho et al., (2022)  Large language models are reasoning teachers \
> [2] Shridhar et al., (2022) Distilling multi-step reasoning capabilities of large language models into smaller models via semantic decompositions \
> [3] Wang et al., (2022) Pinto: Faithful language reasoning using prompt-generated rationales \
> [4] Li et al., (2022)  Explanations from large language models make small reasoners better \
> [5] Touvron et al. (2023) LLaMA: Open and Efficient Foundation Language Models \
> [6] Zhang et al.(2023) AlpaCare:Instruction-tuned Large Language Models for Medical Application \
> [7] Wu et al. (2023)  PMC-LLaMA: Towards Building Open-source Language Models for Medicine \
> [8] Li et al (2023) ChatDoctor: A Medical Chat Model Fine-Tuned on a Large Language Model Meta-AI (LLaMA) Using Medical Domain Knowledge \
> [9] Han et al (2023) MedAlpaca -- An Open-Source Collection of Medical Conversational AI Models and Training Data \
> [10] Xu et al (2023) Baize: An Open-Source Chat Model with Parameter-Efficient Tuning on Self-Chat Data \
> [11]  Gao et Al. (2021) A framework for few-shot language model evaluation

---

> > ### Comment · Reviewer_TtE2 · 2023-11-21
> > **Thanks for replies and follow-up questions**
> >
> > > We define the privacy budget as the ratio of the number of words from the original question that are provided to the third party for prompting the language model
> >
> > I am still concerned that the words from the original question could include PIIs, for example, names. How does **the proposed method** avoid the situation, theoretically or empirically? If the method cannot limit the PII leakage, the claim of 'privacy-preserving' will be concerned.
> >
> > > Concerns about motivation to utilize ChatGPT instead of open-source LLMs.
> >
> > Thanks for your replies. My concern about motivation is addressed.

---

> > > ### Author Response · Authors · 2023-11-22
> > >
> > > Thank you for your continued engagement.
> > >
> > > In practice, we can employ additional de-identification models and rules to in the preprocessing or postprocessing stages to avoid the PII information leakage. Additionally, we can involve a human review to double-check the data before it is transmitted to an API for practical applications.
> > >
> > > The concept of 'privacy-preserving' is crucial because actual hospital environments prohibit sharing even anonymized data with external entities [1]. Our privacy budget is designed in accordance with this policy, determining the proportion of words from the initial query that prompting medical knowledge from the third party LLMs.
> > >
> > >
> > > [1] Johnson et al. (2016). MIMIC-III Clinical Database.

---

> > > > ### Comment · Reviewer_TtE2 · 2023-11-22
> > > > **Thank you**
> > > >
> > > > I strongly suggest authors add experimental results to demonstrate the privacy protection using the mentioned methods.
> > > >
> > > > Without some concerns of lacking empirical support, I think the work can be improved and has value for the community.
> > > > Therefore, i raise my score.

---

### Official Review · Reviewer_EBQC · 2023-11-05

**Soundness:** 3 good
**Presentation:** 3 good
**Contribution:** 3 good
**Rating:** 6
**Confidence:** 3

**Summary:**

In situations where text data is subject to privacy protection constraints, this paper designs a small-scale language model to perform diagnoses of diseases. Utilizing the rich prior medical knowledge in LLM, the approach involves generating a medical knowledge-intensive context using privacy-protected text. This generated context, along with key terms extracted from the text and questions, is then input into the SLM, which is fine-tuned during training. Experiments across multiple datasets demonstrate that this fine-tuning process effectively enhances the accuracy of the diagnostic model.

**Strengths:**

1. This paper focuses on a very important research topic in the field of medicine: how to effectively extract more useful information from incomplete text under the conditions of privacy protection. The author has made full use of the domain knowledge in LLM to effectively fine-tune the SLM, which ensures that the lightweight models can achieve high accuracy.

2. This paper presents rich and comprehensive experiments. Beyond basic decision-making tasks, it also explores solutions for few-shot experiments and out-of-distribution (OOD) model generalization using the methods discussed in this paper.

3. This paper fully utilizes the rich domain knowledge in LLMs to expand the knowledge base of medical reports, achieving excellent diagnostic accuracy even while ensuring privacy protection.

**Weaknesses:**

1. The contribution of this paper to the algorithm and the significance of the clinical problems it addresses seem not to be very high.

2. The main work of this paper appears more as an engineering problem, transferring domain knowledge from LLMs to SLMs. From the perspective of algorithmic contribution, there seems to be some room for improvement.

**Questions:**

1. The experimental datasets in this paper are all question-and-answer test datasets, and whether the methods of this paper are applicable to medical report datasets requires additional experimentation. This is because in medical reports, how to generate high-quality questions using other LLM interfaces is a question worth studying.

2. Large language models provide additional domain knowledge, but in the context of specific medical tasks, will the direct transfer of knowledge from LLMs to SLMs lead to incorrect information leakage into SLMs? How can we ensure that LLMs only enhance information relevant to the current medical issue without introducing additional errors or irrelevant information? This is a very important issue in the medical field, as it directly relates to patient diagnosis.

---

> ### Author Response · Authors · 2023-11-18
>
> We would like to thank the reviewer for their insightful comments and feedback. Their positive remarks regarding motivation, comprehensive experiments, and our efforts in addressing privacy concerns are truly encouraging. Please see our response to your concerns.
>
> **1. Concerns about contribution in the clinical domain.**
>
> The reviewer asserts that there seems to be room for improvement in our algorithm, but it is not clear to us from which perspective we should improve. In addition, we wish to highlight that the contributions of a paper can be evaluated from various perspectives, not limited to algorithms alone.
>
> The primary contribution of our paper is the proposal of a simple yet effective method to utilize LLM as a medical knowledge database, which enhances the decision-making abilities of downstream SLMs. Unlike previous works in medical QA that utilize more complex algorithms and extensive engineering to integrate medical knowledge graphs or databases [1-3], our approach simplifies this by foregoing the complex engineering involved in building knowledge graphs and training algorithms such as GNNs. It has been empirically validated to outperform these methods, as demonstrated in Table 1 in the paper.
>
> Furthermore, our work pioneers the discussion on privacy concerns within the utilization of LLMs in the medical field, a critical issue that has been overlooked in previous research. This innovation opens up new pathways for the deployment of LLMs in sensitive domains.
>
> We also note that the reviewer has acknowledged our contributions to the medical domain with comprehensive experiments and our efforts to mitigate privacy concerns in the review of the strengths section. In this case, we kindly request the reviewer reevaluates our paper based on the clear contribution.
>
> **2. Concerns about experiments in medical examination QA datasets.**
>
> In this paper, our primary focus is on multiple-choice medical QA instead of question generation to evaluate the effectiveness of our approach for several reasons. Firstly, these datasets encompass a wide range of medical topics, including anesthesia, radiology, and pharmacology [4], and different medical applications such as diagnoses and examinations [5]. We can comprehensively evaluate the performance of our methods across various crucial medical domains. Secondly, evaluating question generation tasks presents significant challenges due to limited datasets and the high cost of performing real-world evaluations with clinicians. Hence, we chose medical QA datasets to strike a balance between evaluating medical ability and feasibility, following previous works [1-3,6-9]. We will add the reason for choosing medical examinations in our revision for further clarification.
>
> **3. Concerns about medical knowledge generation methods and denoise capacity of SML.**
>
> Although LLM cannot always generate perfect medical knowledge and may produce noise, alternative external knowledge auxiliary methods are also imperfect.  For example, retrieving information from knowledge graphs and bases not only requires a separate retriever, but also yields noisy and incomplete results since these sources are human-constructed. Consequently, retrieval processes might acquire irrelevant knowledge, potentially hurting model performance.
>
> Additionally, We show that  SLM possesses robust denoising capabilities in Section 5.  Even when received with partially correct noisy medical knowledge, our SLM can  make a correct decision by extracting useful information.  Furthermore, other potential methods, such as integrating LLM with the internet,  could improve the generation quality of LLM for more reliable medical knowledge.
>
>
> [1] Yasunaga et al. (2022a) Qa-gnn: Reasoning with language models and knowledge graphs for question answering \
> [2] Zhang et al. (2022) Greaselm: Graph reasoning enhanced language models for question answering \
> [3] Yasunaga et al. (2022b) Deep bidirectional language-knowledge graph pretraining \
> [4] Pal et al. (2022) MedMCQA : A Large-scale Multi-Subject Multi-Choice Dataset for Medical domain Question Answering \
> [5] Jin et al. (2020)  What Disease does this Patient Have? A Large-scale Open Domain Question Answering Dataset from Medical Exams \
> [6] Mao et al. (2022) Hierarchical representation-based dynamic reasoning network for biomedical question answering \
> [7] Liu et al. (2020)  Interpretable multi-step reasoning with knowledge extraction on complex healthcare question answering \
> [8] Dai et al., (2022) Mixture of experts for biomedical question answering \
> [9] Liévin et al., (2022) Variational open-domain question answering

---

### Author Response · Authors · 2023-11-23
**Kindly Request for Final Review and Feedback**

Dear Reviewers,

We extend our deepest thanks for your valuable feedback. In response to your concerns, we have provide detailed explanation and conducted additional experiments to fortify our study. As the discussion period is concluding, we kindly request that you review our updated responses and please let us know if you have any additional concerns.

Thank you,
authors

---

### Meta-Review · Area_Chair_5db5 · 2023-12-10

**Metareview:**

The paper presents an approach to enhance the performance of small language models (SLMs) in medical diagnostic tasks while preserving privacy. The authors propose a method where keywords are extracted from medical texts and used to generate context through large language models (LLMs), which is then fed into SLMs. This approach aims to leverage the extensive medical knowledge embedded in LLMs while ensuring that the privacy of the text data is maintained. The paper includes experiments across multiple datasets, demonstrating improvements in diagnostic accuracy.

The reviewers generally agree that the paper addresses an important topic in medical AI, focusing on leveraging LLMs to enhance SLMs under privacy constraints. The strengths highlighted include the novelty of the application, the comprehensive experimental evaluation, and the significant improvements in prediction accuracy over baseline methods.

There is a concern regarding the specific definition of private information and how the proposed method ensures privacy protection. Reviewers pointed out potential risks in the keyword extraction process, which might inadvertently include private identifiable information (PII). The paper lacks a theoretical or empirical evaluation of privacy leakage rates. Some reviewers expressed concerns about the overall innovation of the methodology, suggesting that the paper relies heavily on existing methods, such as medical NER models. The contribution to algorithmic development in the field seems limited, and the paper is perceived more as addressing an engineering problem. Questions were raised about the applicability of the method to different types of medical datasets, such as medical reports, and how the approach would handle the generation of high-quality questions in these contexts. There is a query regarding how the proposed method deals with potentially long concatenated inputs, which might exceed the word window of SLMs. The motivation for using ChatGPT over open-source LLMs was initially unclear, although this concern appears to have been addressed in the authors' response.

While the paper presents a valuable contribution to the field of medical AI, especially in terms of privacy-preserving techniques and leveraging LLMs for enhancing SLMs, there are areas that require further clarification and improvement. These include a more rigorous evaluation of privacy protection, clearer definition and handling of private information, and an exploration of the method's applicability to a broader range of medical data types. Additionally, addressing the concerns about the methodological innovation and providing more detailed comparisons with other LLMs could strengthen the paper.

**Justification For Why Not Higher Score:**

There is a concern regarding the specific definition of private information and how the proposed method ensures privacy protection. Reviewers pointed out potential risks in the keyword extraction process, which might inadvertently include private identifiable information (PII). The paper lacks a theoretical or empirical evaluation of privacy leakage rates. Some reviewers expressed concerns about the overall innovation of the methodology, suggesting that the paper relies heavily on existing methods, such as medical NER models. The contribution to algorithmic development in the field seems limited, and the paper is perceived more as addressing an engineering problem. Questions were raised about the applicability of the method to different types of medical datasets, such as medical reports, and how the approach would handle the generation of high-quality questions in these contexts. There is a query regarding how the proposed method deals with potentially long concatenated inputs, which might exceed the word window of SLMs. The motivation for using ChatGPT over open-source LLMs was initially unclear, although this concern appears to have been addressed in the authors' response.

**Justification For Why Not Lower Score:**

The reviewers generally agree that the paper addresses an important topic in medical AI, focusing on leveraging LLMs to enhance SLMs under privacy constraints. The strengths highlighted include the novelty of the application, the comprehensive experimental evaluation, and the significant improvements in prediction accuracy over baseline methods.

---

### Decision · Program_Chairs · 2024-01-16

Accept (poster)